# Computational Mechanisms of Approach-Avoidance Conflict Predictively Differentiate Between Affective and Substance Use Disorders

**MARISHKA M. MEHTA** (ID)

**NAVID HAKIMI** (ID)

**ORESTES PENA** (ID)

**TAYLOR TORRES** (ID)

**CARTER M. GOLDMAN** (ID)

**CLAIRE A. LAVALLEY** (ID)

**JENNIFER L. STEWART** (ID)

**HANNAH BERG** (ID)

**MARIA IRONSIDE** (ID)

**MARTIN P. PAULUS** (ID)

**ROBIN AUPPERLE** (ID)

**RYAN SMITH** (ID)

*Author affiliations can be found in the back matter of this article

**CORRESPONDING AUTHOR:**
**Ryan Smith**

*Laureate Institute for Brain Research. 6655 S. Yale Ave. Tulsa, Oklahoma, 74136, US*

rsmith@laureateinstitute.org

**KEYWORDS:**
Depression; Anxiety; Substance Use Disorders; Computational Modeling; Predictive Classification; Approach-Avoidance Conflict

**TO CITE THIS ARTICLE:**
Mehta, M. M., Hakimi, N., Pena, O., Torres, T., Goldman, C. M., Lavalley, C. A., Stewart, J. L., Berg, H., Ironside, M., Paulus, M. P., Aupperle, R., & Smith, R. (2025). Computational Mechanisms of Approach-Avoidance Conflict Predictively Differentiate Between Affective and Substance Use Disorders. *Computational Psychiatry*, 9(1), pp. 159–186. DOI: https://doi.org/10.5334/cpsy.131

## ABSTRACT

Psychiatric disorders are highly heterogeneous and often co-morbid, posing specific challenges for effective treatment. Recently, computational modeling has emerged as a promising approach for characterizing sources of this heterogeneity, which could potentially aid in clinical differentiation. In this study, we tested whether computational mechanisms of decision-making under approach-avoidance conflict (AAC) – where behavior is expected to have both positive and negative outcomes – may have utility in this regard. We first carried out a set of pre-registered modeling analyses in a sample of 480 individuals who completed an established AAC task. These analyses aimed to replicate cross-sectional and longitudinal results from a prior dataset (N = 478) – suggesting that mechanisms of decision uncertainty (*DU*) and emotion conflict (*EC*) differentiate individuals with depression, anxiety, substance use disorders, and healthy comparisons. We then combined the prior and current datasets and employed a stacked machine learning approach to assess whether these computational measures could successfully perform out-of-sample classification between diagnostic groups. This revealed above-chance differentiation between affective and substance use disorders (balanced accuracy > 0.688), both in the presence and absence of co-morbidities. These results demonstrate the predictive utility of computational measures in characterizing distinct mechanisms of psychopathology and may point to novel treatment targets.

The presence of comorbid psychiatric disorders is associated with greater severity, chronicity, poorer treatment response, and overall functional impairment (Kessler et al., 2015; Klein Hofmeijer-Sevink et al., 2012; Plana-Ripoll et al., 2020; Plana-Ripoll et al., 2019). Comorbid psychiatric disorders show considerable heterogeneity, which can pose further difficulties for prognosis and treatment. Unfortunately, the mechanisms leading to this pattern of comorbidity and heterogeneity remain poorly understood, limiting the development of more targeted treatments. It is expected that some of the mechanisms underlying psychiatric disorders are transdiagnostic and account for comorbidities (Dalgleish et al., 2020), while other mechanisms may differentiate clinical profiles with distinct underlying causes. If so, developing more targeted and individualized treatments will likely require identification of both transdiagnostic and diagnosis-specific mechanisms.

One prominent set of cognitive and neurocomputational mechanisms of transdiagnostic relevance pertains to approach-avoidance conflict (AAC), where available choices are expected to lead to both positive and negative outcomes (Aupperle & Paulus, 2010). In depression and anxiety disorders (DEP/ANX), for example, maladaptive resolution of this conflict can manifest as avoidance of rewarding activities in anticipation of feared negative outcomes (Barlow et al., 2016). Similarly, individuals with substance use disorders (SUDs) often experience conflict between cravings, withdrawal, and the negative consequences of dependence, possibly leading to continued substance use despite loss of career, social support, and physical health. Given its broad relevance, a growing body of clinical research has emerged in recent years examining maladaptive decision processes under AAC in different clinical populations (for a detailed review, see Letkiewicz et al. (2023)), with the goal of identifying novel and individualized treatment targets (Paulus, 2017). Within this body of work, several paradigms have been used to study AAC (reviewed by Kirlic et al., 2017). More recently, computational modeling approaches have highlighted cognitive processes, such as suboptimal reward valuation, uncertainty, or inference, that may contribute to maladaptive AAC behavior. However, the clinical utility of these findings remains limited by sparse evidence of replicability and longitudinal stability of computational markers, as well as indications of variability in test-retest reliability (ranging from poor-to-excellent) depending on the task in question (Brown et al., 2020; Chung et al., 2017; Enkavi et al., 2019; Karvelis et al., 2023; Moutoussis et al., 2019; Price et al., 2019; Shahar et al., 2019). This limitation partly arises from the prevalent use of cross-sectional study designs, though more recent shifts toward longitudinal methodologies have facilitated identification of computational markers demonstrating good reliability (Goodwin et al., 2024). However, these investigations have predominantly focused on relatively short intervals spanning days to weeks. Investigations focusing on long-term stability will be crucial for progress in computational psychiatry and its continued efforts to identify underlying mechanisms and predictors of treatment outcomes. In previous work, we have investigated cross-sectional differences in, and longitudinal stability of, AAC behavior in transdiagnostic samples including healthy comparisons (HCs), individuals with DEP/ANX, and individuals with SUDs (Smith et al., 2021b; Smith et al., 2021c). This work made use of a novel computational model capable of disentangling two dimensions of AAC: the subjective value of negative stimuli relative to reward, referred to as emotion conflict (*EC*), and the degree to which individuals are unsure about the best choice, referred to as decision uncertainty (*DU*). We found that individuals in both clinical groups showed higher *DU* and lower *EC* than HCs, which was consistent at a 1-year follow-up visit. The cross-sectional results at baseline were also recently replicated in an independent sample (Smith et al., 2023), suggesting generalizability. However, the longitudinal findings remain to be replicated. It also remains unclear whether these model-based measures support out-of-sample prediction and whether they are fully transdiagnostic or might also differentiate clinical sub-populations.

Our previous work also found that observed group differences in *EC* were driven largely by females. This added to a growing literature on sex differences in diagnostic frequency, symptom experience, and coping mechanisms (Gobinath et al., 2017; Green et al., 2019; Kelly et al., 2008; Matud, 2004; Riecher-Rössler, 2017; Rutter et al., 2003). For example, females appear more prone to affective disorders (Green et al., 2019; Rutter et al., 2003), and specific difficulties in emotion regulation have also been associated with symptom severity in females but not males (Kelly et al., 2008). It is therefore possible that the sex differences we have observed in AAC behavior could contribute to these specific difficulties.

The current study aimed to address three central questions. First, we tested the replicability of previously reported longitudinal results and observed sex differences in a new community-based sample. These pre-registered analyses (https://osf.io/7hsx9) specifically aimed to replicate: (1a) previously observed reductions in *DU* over time; as well as (1b) consistently elevated *DU* over time in both DEP/ANX and SUDs relative to HCs. We further aimed to replicate: (1c) previously observed reductions in *EC* within each clinical group compared to HCs that were stable over time, where this effect was driven by differences in females but not males. After completing each of these pre-registered analyses, we then combined the prior and current datasets, which afforded greater statistical power to test questions that we were limited in our ability to explore previously. This included examining: (2) whether observed differences by group and sex were driven by narrower diagnostic categories (i.e., for each specific affective and substance use disorder); and (3) whether computational measures afforded predictive utility in distinguishing between diagnoses. To do so, we used state-of-the-art machine learning approaches to test the out-of-sample predictive accuracy of these computational measures in classifying individuals into DEP/ANX or SUD groups. This provided a direct test of whether *DU* and *EC* levels could jointly contribute to differential diagnosis (i.e., differentiating those with only an affective disorder, the presence of at least one SUD, or both). Together, these analyses aimed to demonstrate whether a computational characterization of behavior under AAC might provide both mechanistic and pragmatic information that could be useful in a clinical setting.

## METHODS

### PARTICIPANTS

This study leveraged a community-based sample (N = 1050) from the Tulsa 1000 (T1000) study (Victor et al., 2018), which was recruited through radio, electronic media, treatment center referrals, and word of mouth. All participants were between the ages of 18–55 years. This study recruited both healthy comparisons (HCs) and a transdiagnostic clinical group. Within the latter group, individuals with SUDs were recruited from community recovery homes and were currently abstinent. Recruitment for the clinical group began with a transdiagnostic population screening, where participants were required to meet symptom severity cutoffs on at least one of the following dimensional measures: the Patient Health Questionnaire-9 (i.e., depression severity [PHQ]; Kroenke et al., 2001) ≥ 10, the Overall Anxiety Severity and Impairment Scale (OASIS; Norman et al., 2006) ≥ 8, and the Drug Abuse Screening Test (DAST-10; Bohn et al., 1991) ≥ 3. For individuals who met one or more of these initial cutoffs, an in-depth diagnostic interview was performed at baseline in accordance with DSM-IV or DSM-5 criteria using the Mini International Neuropsychiatric Inventory (MINI; Sheehan et al., 1998), administered at the baseline visit by a trained professional. Participant diagnostic groupings into those with depression, anxiety, and/or substance use disorders (based on the MINI) at baseline were retained at 1-year follow-up, irrespective of change in self-reported symptoms (while analyses assessing change in symptoms over time tested for relevant potential effects; see **Supplementary Figure 1**). HCs did not show elevated symptoms or meet criteria for any psychiatric diagnosis. Exclusion criteria included a positive test for drugs of abuse; diagnosis of psychotic, bipolar, or obsessive-compulsive disorders; or reported history of moderate to severe traumatic brain injury, neurologic disorders, or severe or unstable medical conditions, active suicidal intent, or plan, or change in medication dose within 6 weeks. For detailed inclusion and exclusion criteria, see Victor et al. (2018).

As in our previous work in an *exploratory* sample (i.e., taken from the first 500 participants recruited to the T1000 study; Smith et al., 2021b; Smith et al., 2021c), the *confirmatory* sample (i.e., taken from the subsequent 550 participants recruited to the T1000 study; Smith et al., 2023) was divided into three groups: (i) HCs (baseline: N = 97, 1-year follow-up: N = 69); (ii) DEP/ANX (baseline: N = 208, 1-year follow-up: N = 135; including individuals who met criteria for one or more of the following: major depressive disorder, social anxiety disorder, generalized anxiety disorder, panic disorder, and/or posttraumatic stress disorder); (iii) SUDs (baseline: N = 175, 1-year follow-up: N = 83; with or without comorbid anxiety/depression). A full breakdown of co-morbidities within the clinical

sample (i.e., individuals who met criteria for DEP/ANX or SUDs) is provided in **Supplementary Table 1**. Here it is also worth highlighting that symptom cutoffs and diagnostic groupings did not always perfectly align, as individuals who met criteria for a given disorder (e.g., depression) may not have indicated elevated symptoms for that disorder on self-report measures (e.g., PHQ < 10), but nonetheless were screened into the study because they met the cutoff for another measure (e.g., OASIS ≥ 8).

## DATA COLLECTION PROCEDURE

The T1000 protocol included intensive assessment of demographic, clinical, and psychiatric features, with a focus on negative and positive affect, arousal, and cognitive functioning. The complete list of assessments, along with their validity and reliability, are reported elsewhere (Victor et al., 2018). Here, we focus mainly on the symptom measures gathered at screening (PHQ, OASIS and DAST-10). However, our previous study did also explore potential associations with some of the other available assessments (Smith et al., 2021c). Thus, we also aimed to replicate those results here. These secondary measures included: the Patient-Reported Outcomes Measurement Information System (PROMIS) depression and anxiety scales (Cella et al., 2010), the Behavioral Activation/Inhibition scales (BIS/BAS; Carver & White, 1994), the Positive and Negative Affect Schedule (PANAS; Watson et al., 1988), the Anxiety Sensitivity Index (ASI; Sandin et al., 2001), the Temporal Experience of Pleasure Scale (TEPS; Gard et al., 2006), and the State-Trait Anxiety Inventory (STAI; Spielberger et al., 1970). Use of these dimensional measures aligned with the broader goals of the T1000 project, which was designed around the NIMH Research Domain Criteria (RDoC). All data collection procedures were approved by the Western Institutional Review Board. All participants provided written informed consent before completion of the study protocol, in accordance with the Declaration of Helsinki, and were compensated for participation (ClinicalTrials.gov identifier: #NCT02450240).

## APPROACH-AVOIDANCE CONFLICT (AAC) TASK

The AAC task (Figure 1) used here has been described in detail in our previous work (Aupperle et al., 2015; Aupperle et al., 2011; Smith et al., 2021b). Briefly, a picture of an avatar standing above a runway, at its starting position, was presented on each trial. Participants were asked to move the avatar to one of the nine positions on the runway, toward the cues presented on either side. The cue consisted of an image – sun or cloud – representing a positive or negative affective image-sound pair, respectively, that would be shown with higher probability as the avatar moved closer to the associated side (detailed below). There was also a rectangular bar on each side, where the height of the red fill represented reward points associated with the linked image-sound pair. There were five trial types across 60 trials (12 each), defined by the cues presented: avoid-threat (AV), approach-reward (APP), and three levels of conflict trials (CONF2, CONF4, CONF6). In AV trials, the cloud and sun images were presented on opposing sides, with 0 reward points associated with either. APP trials provided positive affective stimuli on both sides, where one side was associated with 0 points and the other with 2 points. Lastly, in conflict trials, cues presented always included the sun with 0 reward points and the cloud with levels of 2 (CONF2), 4 (CONF4), and 6 (CONF6) reward points. These reward points did not lead to additional monetary compensation. The affective images and sounds were sampled from the International Affective Picture System (IAPS; Lang et al., 2008), International Affective Digitized Sounds (IADS; Lang & Bradley, 1999), and other freely available audio files (refer to Aupperle et al. (2015) and Chrysikou et al. (2017)). Some examples of visual stimuli include images depicting violence or suffering, scenic natural environments, and people displaying positive emotions (e.g., smiling or laughing) or engaging in joyful activities. Examples of associated auditory stimuli include screaming noises, the sound of nails on a blackboard, wind bells, and pleasant laughter, to name a few.

Here it should be noted that the AAC task used in the T1000 studies differs in a few respects from the original version described by Aupperle et al. (2011). First, the original task did not include the APP condition, which was introduced in a later adaptation by Aupperle et al. (2015), and has been used in all subsequent versions of the task. Second, the T1000 version included a reduced number

Mehta et al.                      **163**
*Computational Psychiatry*

of trials (60 total; 12 trials per condition), compared to the 90-trial format used in earlier versions of the task. Due to this shorter design, the avatar's starting position was also restricted to either the center or the end of the runway (left or right), counterbalanced across conditions. In contrast, the 90-trial version allowed the avatar to begin from any position on the runway, permitting two trials per starting position within each condition.

Model-free task measures included: 1) average chosen runway position; 2) choice variability, measured as the within-subject standard deviation in chosen runway position (i.e., measuring response consistency); and 3) response times (RTs, i.e., time to initial button press) across trials. These measures were calculated both as overall averages across all trial types and separately within each task condition. In addition, participants completed a short post-task Likert scale survey (specific questions detailed below).

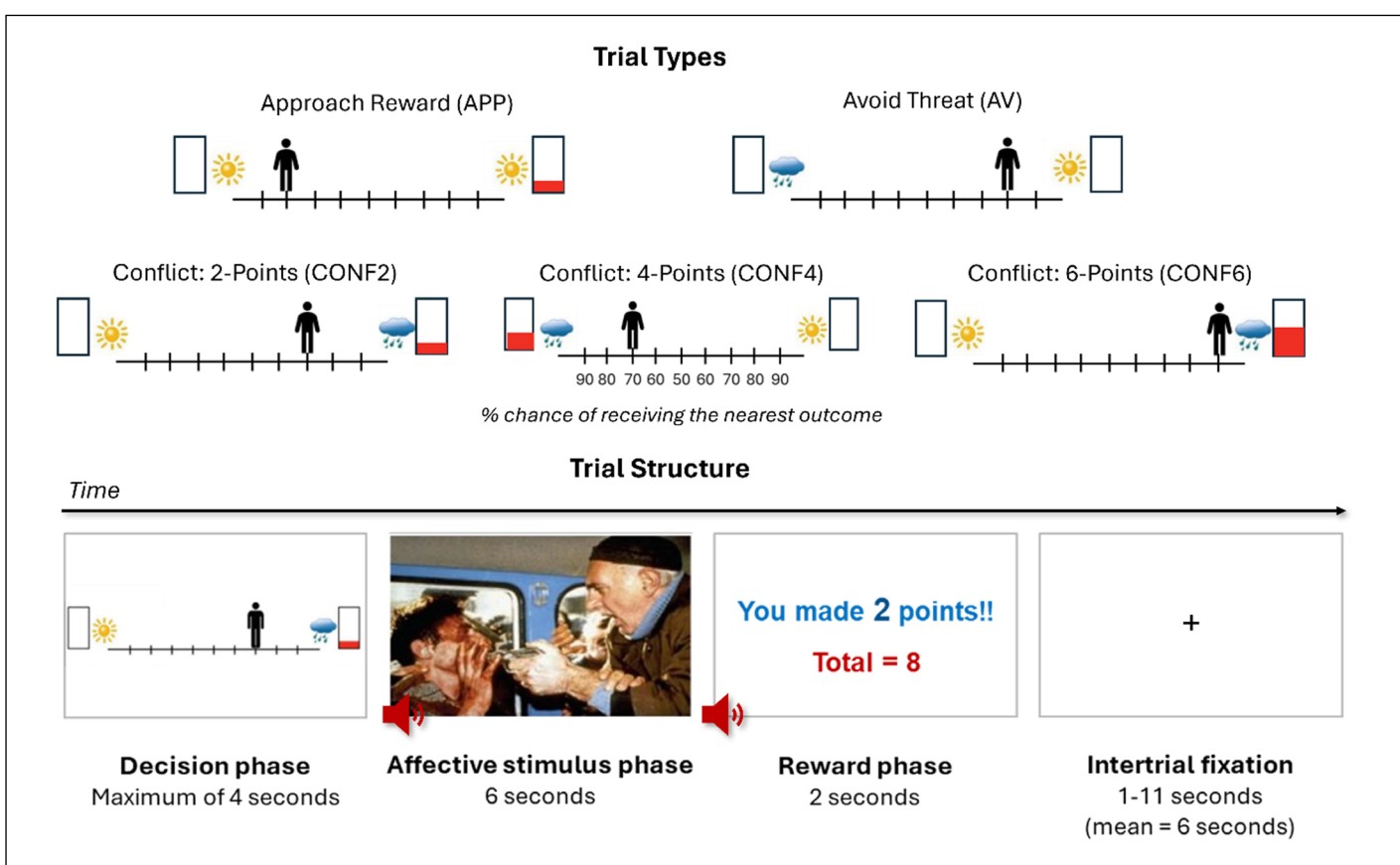

**Figure 1 The approach-avoidance conflict (AAC) task.** *Bottom*: Each trial is divided into a decision phase, an affective stimulus phase, and a reward phase. Trials are separated by a variable intertrial fixation time. *Top*: During the decision phase, participants choose to move an avatar to one of nine positions on a runway. Pictures are presented on each side of the runway, indicating the types of stimuli that could be presented during the affective stimulus and reward phases. The sun and cloud images represented potential positive and negative affective stimuli, respectively (each being an image–sound combination). The height of the red fill in a rectangle signified the number of points that would be received in the reward phase (ranging from 0 to 6 points). Participants were instructed that the final position of the avatar determined the probability of each of these outcomes occurring (in increments of 10%, from 90% to 10% with each step away from the associated stimulus indicator images). All choices therefore resulted in a probabilistic outcome. For example, if a participant chose the highest probability option on a given side (i.e., the runway position closest to their preferred outcome), there was a 90% chance that the preferred outcome would be presented. However, there was still a 10% chance that the non-preferred outcome associated with the alternative side of the runway would be presented instead. In the CONF6 condition above, for instance, the preferred outcome might be the combination of a negative affective stimulus and 6 points, while the alternative outcome would be a pleasant affective stimulus associated with no points. At the end of each trial (i.e., in the reward phase), participants were informed of the points won on that trial (i.e., including when 0 points were earned) as well as the total number of points they had acquired in the task thus far. The five trial types and associated probabilities of each outcome at each runway position are also shown above. The task consisted of 60 trials, with 12 of each of the five trial types.

## COMPUTATIONAL MODELING

As in our previous studies (Smith et al., 2021b; Smith et al., 2021c; Smith et al., 2023), a two-parameter partially observable Markov decision process (POMDP) framework (Table 1) was used to model AAC task behavior. The model estimated parameters reflecting decision uncertainty (DU) and the subjective aversion to negative stimuli (i.e., relative to subjective value assigned to each point that could be won; "emotion conflict" [EC]). Higher EC values reflect greater avoidance of negative image-sound pairs, depending on the reward point value on offer. Higher DU values suggest greater variability in choice under identical task conditions and the tendency to adopt choices nearer to the middle of the runway. Thus, these parameters capture two dimensions of AAC.

Mehta et al. 164
*Computational Psychiatry*

**Table 1** Markov decision process model of the AAC task.

**Note:** DU = decision uncertainty; EC = emotion conflict. Trial types: Approach (APP), Avoid (AV), Conflict (2: CONF2, 4: CONF4, 6: CONF6).

| MODEL VARIABLE | GENERAL DEFINITION | MODEL-SPECIFIC SPECIFICATION |
|---|---|---|
| $o_t$ | Observable outcomes at time $t$. | **1.** Observed position on the runway (10 possible observations. This included a "starting" position and the nine final positions on the runway that could be chosen). |
| | | **2.** Cues indicating trial type (five possible observations, corresponding to the five trial types). |
| | | **3.** Stimuli observed at the end of each trial. This formally included seven possible observations: a "starting" observation, the positive stimulus with 0 or 2 points, and the negative affective stimulus with 0, 2, 4, or 6 points. |
| $s_t$ | Beliefs about hidden states at time $t$. | **1.** Beliefs about position on the runway: 10 possible belief states with an identity mapping to the observations in $o_t$ (runway position). |
| | | **2.** Beliefs about trial type: 5 possible belief states (corresponding to the five trial types) with an identity mapping to the observations in $o_t$ (trial types). |
| $\pi$ | A distribution over action options encoding the probability of choosing each action. | Encodes all the allowable actions. Here, actions are decisions to transition from the starting state to each of the nine possible runway positions. |
| $\beta$ (**DU**) | A 'temperature' parameter within a softmax choice function. Higher values for this parameter reduce the likelihood that actions expected to generate the most preferred outcomes will be chosen. | Higher **DU** values reflect lower confidence about the best decision and lead to less deterministic action selection. This means that a participant is more likely to choose locations in the middle of the runway and more likely to make inconsistent choices across trials. Therefore, **DU** encodes participants' decision uncertainty during the task. |
| $p(o_t\|s_t)$ | A matrix encoding beliefs about the relationship between hidden states and observable outcomes (i.e., the likelihood that specific outcomes will be observed given specific hidden states). | Encodes beliefs about the relationship between position on the runway and the probability of observing each outcome, given beliefs about trial type. |
| $p(s_{t+1}\|s_t, \pi)$ | A matrix encoding transition probabilities, or beliefs about how hidden states will evolve over time depending on choice of action. | Encodes beliefs about all the possible ways participants could choose to move the avatar, along with the belief that the task condition (trial type) will not change within a trial. |
| $\ln p(o_t\|C)$ | A matrix **C** encodes the degree to which some observed outcomes are preferred over others at each time point $t$ (technically modeled as prior expectations conditioned on subjective stimulus value). The values for each column (representing $t$) in this matrix are passed through a softmax function to generate a probability distribution, which is then log-transformed. | Encodes stronger positive preferences for receiving higher amounts of points, and negative preferences for the aversive stimuli. These values are relative to an anchor value of 0 (for the "safe" positive stimulus). The **EC** parameter encodes the value of participants' preferences against observing the aversive stimuli (i.e., prior to being passed through the softmax function and log-transformed). This value therefore competes against the value of points that could be won (e.g., if EC = 4, this means at least four reward points are needed to outweigh the unpleasantness of the aversive stimuli). |
| $p(s_1)$ | A matrix encoding beliefs about (a probability distribution over) initial hidden states. | The simulated participant always begins in an initial starting state and believes that each trial type is equally likely (before observing the trial type cues). |

Parameter estimates were optimized by maximizing the likelihood of participants' choice behavior using a standard Bayesian approach called Variational Laplace (Friston et al., 2007). Model simulations were done using the spm_MDP_VB_X.m MATLAB script, within the freely available SPM academic software package (http://www.fil.ion.ucl.ac.uk/spm/). MATLAB scripts used to model the task are provided in Appendix 3 of Smith et al. (2021b).

## INITIAL ASSESSMENT OF SAMPLE DIFFERENCES

Differences in participant characteristics were measured at the baseline and 1-year follow-up visit, and between the exploratory and confirmatory samples using frequentist and Bayesian approaches. For each group, possible differences in model parameters at baseline, follow-up, and their pre-post change were separately assessed using *t*-tests and linear regressions. Equivalent Bayesian *t*-tests and regressions were performed to compute Bayes Factors (BFs; using the *BayesFactor* package in R [https://richarddmorey.github.io/BayesFactor/]), which reflect the ratio of the probability of observed data under models with vs. without an effect of interest. Thus, BFs < 1 provide evidence in favor of a null model. For parameter change scores, these Bayesian regression models included sample type and baseline scores as predictors of change scores. The null model retained baseline scores as a predictor to isolate the effect of sample type (exploratory vs. confirmatory) from any potential confounding effects of baseline differences.

## PRE-REGISTERED STATISTICAL ANALYSIS IN CONFIRMATORY SAMPLE

All statistical analyses were performed using the R statistical package (R Core Team, 2023). Each analysis below follows the pre-registered analysis plan (https://osf.io/7hsx9) for replicating our prior results (Smith et al., 2021c). As detailed further below, additional analyses were subsequently conducted when combining the exploratory and confirmatory datasets to address more granular clinical group differences and out-of-sample classification of those with each disorder.

### Model validity

Model performance was evaluated using the average action probability and accuracy scores under the best-fit model for each participant, reflecting the average percentage of trials for which the action with the highest probability in the model matched the action chosen. The relationship between model parameters (*EC* and *DU*) and RTs was also examined, with the expectation that higher *EC* and *DU* would correlate with slower RTs. The relationship between parameter estimates and post-task Likert scales was also assessed, with the expectation that higher *EC* values would be associated with greater self-reported avoidance motivation and anxiety during the task, while higher *DU* would be associated with greater self-reported difficulty in making decisions on the task.

### Within-subject stability

Consistency in scores over time (baseline to 1-year follow-up) was evaluated using single-measure consistency intra-class correlations [ICC(3, 1)]. These analyses were carried out for individual parameter estimates (*EC* and *DU*), model-free task measures (RTs, average chosen runway position, choice variability), clinical measures (PHQ, OASIS, and DAST-10), post-task survey items, as well as secondary dimensional measures (ASI, BIS/BAS, PROMIS, PANAS, STAI, and TEPS). This was done across all participants and then further analyzed for each group separately. In line with prior suggestions, ICCs below 0.4 were considered 'poor', greater than 0.40 were considered 'fair', and greater than 0.60 were considered 'good' (Cicchetti, 1994).

### Clinical differences

Linear mixed effects models (LMEs) were used to investigate the impact of group (sum coded; HC = –1) on model parameter estimates, model-free descriptive measures, and post-task scale items. These models also assessed the effect of time (sum coded: baseline = –1; follow-up = 1) and its interaction with group. Further LMEs also confirmed that observed effects were not accounted for by age (centered) or biological sex (sum coded: female = –1; male = 1), along with their interactions with group. Similar to our previous report (Smith et al., 2023), additional LMEs were performed controlling for the effect of Wide-Range Achievement Test Reading Sub-Test (WRAT; a

measure of premorbid cognitive ability; Johnstone et al., 1996) and its interaction with group in a subset of participants with available data ($N_{Baseline} = 391$; $N_{Follow-up} = 238$).

Partial correlations were performed assessing whether baseline task measures predicted change in dimensional measures from baseline to follow-up while accounting for baseline symptom values. Similar analyses were carried out in relation to symptom measures within the clinical groups. Further analyses also tested potential associations between changes in task measures and changes in symptoms over time (accounting for baseline values for each). Bayes factors were also calculated for these correlations to more strongly support null results in cases where frequentist analyses did not replicate previous findings. When computing these Bayes factors, the null hypothesis model included possible effects of baseline symptom levels.

## COMBINED SAMPLE ANALYSES

After completing pre-registered analyses, the exploratory ($N_{Baseline} = 478$; $N_{Follow-up} = 324$) and confirmatory ($N_{Baseline} = 480$; $N_{Follow-up} = 287$) samples were combined to ask novel questions benefiting from additional statistical power.

### Transdiagnostic generalizability

The consistency of clinical differences in model parameters between HCs and more specific diagnostic categories (e.g., generalized anxiety disorder, social anxiety disorder, methamphetamine use disorder, opioid use disorder) within each clinical group was examined using LMEs with the same structure described above: *parameter ~ group*time + (1|participant)* and *parameter ~ group + time + group*age + group*sex + (1|participant)*. Each analysis was performed on a subsample including only the data from HCs and individuals with the specified disorder (irrespective of comorbidities; thus, some participants were included in more than one clinical group). Due to the small sample of individuals with hallucinogen use disorder ($N_{Baseline} = 15$; $N_{Follow-up} = 4$), this subgroup was not analyzed separately.

As some subgroups had only moderate sample sizes, we performed post-hoc power analyses by simulation to examine the influence of these varying sample sizes across specific diagnostic categories. These analyses focused on LMEs with the following structure: *parameter ~ group*time + (1|participant)*. Without losing generality, the between- and within-subject variances were set to 0.1 and 0.9, respectively, such that the total variance is 1. Missing values were introduced to the outcome variable with a 0.25 probability for HCs and between 0.32–0.64 probabilities for each clinical group at 1-year follow-up. These probabilities correspond to the attrition rate for the clinical group in question. Group and visit variables were coded as [–1, 1] such that the associated beta coefficient indicated 1/4 of the mean outcome averaged over visits between groups. Using this setup, the data were then used to fit the aforementioned model structure. This process was repeated 4,000 times, where the computed statistical power represents the proportion of *p*-values < 0.05 for the group term.

### Model-Based Prediction of RTs

A secondary question of interest, which contributed to the machine learning analyses described below, examined whether the choice-based model could capture aspects of decision deliberation reflected by RTs across the combined sample. Here, a trial-by-trial measure of *choice uncertainty* was calculated as the entropy of the probability distribution over actions ($\pi$):

$$Choice\ uncertainity = -\sum_{i=1}^{n} p(\pi_i) \ln p(\pi_i)$$

Pearson correlation coefficients were computed for each participant between trial-by-trial RTs and *choice uncertainty*. An LME was then used to test whether this association was significantly greater than 0 across participants, while controlling for effects of group, time, their interaction, and individual-level random intercepts. Secondary analyses were also performed to evaluate the relationship between RTs and chosen runway position, and RTs and model parameters. Here, LMEs were used to predict chosen runway position and model parameters based on group, time, and RTs, and possible *group × time* and *group × RT* interactions.

## Predictive Utility

As a final aim of the study, we assessed the predictive utility of our computational model parameters. Specifically, we utilized standard binary classification machine learning techniques, applying 5-fold cross-validation to predict group membership within the clinical sample (i.e., removing HCs from all analyses) using computational task metrics. In particular, we tested out-of-sample accuracy in classifying: 1) the exclusive presence of affective disorders or SUDs (i.e., with no comorbidity between the two; N = 320 and 51, respectively); 2) the presence vs. absence of SUDs (with or without comorbid affective disorders; N = 167 and 327, respectively); 3) the presence vs. absence of affective disorders (with or without comorbid SUDs; N = 436 and 58, respectively); and 4) the presence vs. absence of comorbidity generally (i.e., those with only SUDs or only DEP/ANX vs. those with both types of disorders; N = 123 and 371, respectively).

To do so, an established stacked ensemble approach was used (detailed within Ekhtiari et al. (2019)), in which three different classification approaches were selected. Their performance was compared, and an optimized combination of their predictions was then evaluated. The chosen algorithms were selected to represent different general classification approaches: elastic net (ENET), k-nearest neighbors (KNN), and Bagged AdaBoost (ADABAG). Briefly, ENET is a regularized logistic regression approach that combines lasso (L1) and ridge (L2) penalty terms. It is particularly useful in overcoming multicollinearity and avoiding overfitting. A parameter $\alpha$ balances the proportion of the L1 and L2 penalties. A second parameter λ sets the weight on the loss function (parameter optimization is described further below). KNN is a non-parametric supervised learning algorithm. The parameter *k* refers to the number of nearest observations considered to predict the classification label. Finally, ADABAG combines bootstrap aggregation (Bagging) and adaptive boosting (AdaBoost) approaches to improve model accuracy and stability. AdaBoost iteratively classifies weak classifiers (decision trees), where each classifier uses previously misclassified observations. It adaptively increases the weight placed on currently misclassified observations while reducing the weight of data already classified correctly. This aids in reducing bias, variance, and overfitting. However, it can be susceptible to outliers and noisy data. To address this, bagging was used to generate datasets consisting of random samples (with replacement) from the original dataset. AdaBoost models were then trained on these generated datasets. This method improves consistency in prediction by combining the outputs from several weak classifiers, each trained on a generated dataset. AdaBoost has two parameters, *mfinal* and *maxdepth*, which control the number of boosting iterations and maximum tree depth, respectively.

The 5-fold cross-validation consisted of first dividing the dataset into five equal subsets. For each of the five rounds, four subsets were combined and used as a training dataset, and the fifth was used as a testing dataset. In each round, the training dataset was also further divided into training (80%) and validation (20%) subsets. The validation subset was used for finding optimal hyperparameters. Here, seven different starting values were randomly selected for each hyper-parameter, and a grid-search approach was used to find the optimal values. After parameter tuning was complete, the stacked ensemble approach then used linear aggregation to combine the predictions of each algorithm. Specifically, the weight on each algorithm was determined by respective model fit (area under the curve [AUC]), such that algorithms with better fits to the data were afforded higher weight in generating the stacked prediction. This process was repeated five times to get a stable comparison of fitted stacked models (STACK), and we report the average performance across these 5 folds. Following best practices for evaluating models on imbalanced datasets (Thölke et al., 2023), we used AUC and balanced accuracy (arithmetic mean of *sensitivity* and *specificity*) as performance metrics. Previous work using similar approaches in clinical samples is limited, but available studies have found AUCs ranging between 0.7–0.9 and balanced accuracies around 0.7 (Aafjes-van Doorn et al., 2021; Houghton et al., 2023; Lee et al., 2025); thus, we hypothesized similar levels of performance. Sensitivity and specificity, which refer to the true positive and negative rate, respectively, are also reported in results. In addition, a variable importance (VI) metric was computed to measure the average scaled contribution (between 0 to 100) of each predictor. As noted above, due to the unbalanced sample size, an up-sampling approach was also used during the cross-validation process.

Our approach focused on sequentially assessing the degree to which incorporating information derived from our computational model could improve prediction beyond basic demographic variables (age and sex). As such, we tested and compared models with five sets of predictors: 1) age and sex alone; 2) model parameters (*EC* and *DU* at baseline and follow-up) alone; 3) model parameters with age and sex; 4) model parameters and *r* values representing the correlation between trial-by-trial *choice uncertainty* and RTs; and 5) model parameters, *r* values, and age and sex. Because prior and current results show consistent *group × age* and *group × sex* interactions in predicting *DU* and *EC*, inclusion of demographic variables allowed us to test if they were sufficient to predict clinical group membership on their own (i.e., the observed differences are a function of sample characteristics) or whether model-based information led to substantial improvement in classification. All analyses were conducted using the *caretStack* library (https://github.com/kforthman/caretStack) in R. In addition, *glmnet, Matrix, adabag* and *plyr* libraries were used for base models. A full list of available base models can be found at https://topepo.github.io/caret/available-models.html.

Mehta et al.
**168**
*Computational Psychiatry*

## SUMMARY

A schematic summary of the study design, including pre-registered replication aims in the confirmatory 1-year follow-up sample, and subsequent analyses in the combined sample, is presented in Figure 2.

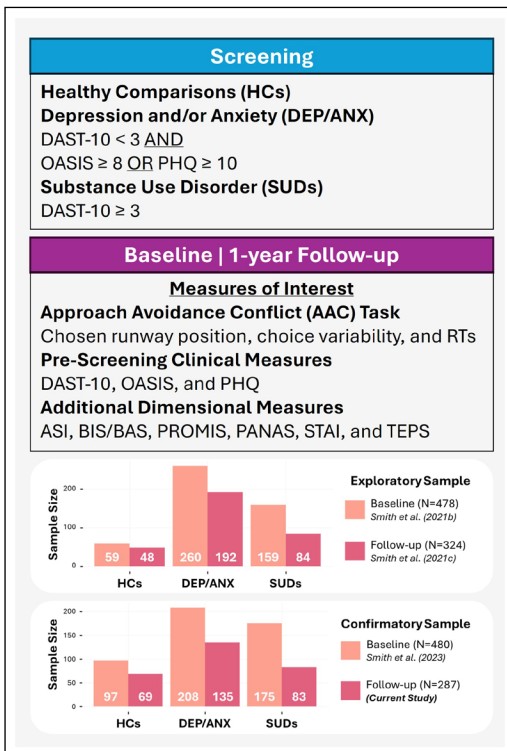

**Screening**

**Healthy Comparisons (HCs)**
**Depression and/or Anxiety (DEP/ANX)**
DAST-10 < 3 AND
OASIS ≥ 8 OR PHQ ≥ 10
**Substance Use Disorder (SUDs)**
DAST-10 ≥ 3

**Baseline | 1-year Follow-up**

**Measures of Interest**
**Approach Avoidance Conflict (AAC) Task**
Chosen runway position, choice variability, and RTs
**Pre-Screening Clinical Measures**
DAST-10, OASIS, and PHQ
**Additional Dimensional Measures**
ASI, BIS/BAS, PROMIS, PANAS, STAI, and TEPS

**Exploratory Sample**
Baseline (N=478) *Smith et al. (2021b)*
Follow-up (N=324) *Smith et al. (2021c)*
HCs: 59, 48 | DEP/ANX: 260, 192 | SUDs: 159, 84

**Confirmatory Sample**
Baseline (N=480) *Smith et al. (2023)*
Follow-up (N=287) *(Current Study)*
HCs: 97, 69 | DEP/ANX: 208, 135 | SUDs: 175, 83

**Pre-registered** aims to **replicate** the long-term stability of 1-year follow-up results in the **confirmatory sample**

**Reliability and validity**
✓ Fair-to-good test–retest reliability of individual parameter estimates
✓ Model parameters remained correlated with reaction times and post-task surveys

**Longitudinal stability**
✓ Reduction in decision uncertainty (*DU*) over time
**X** Group differences in *DU* at follow-up
✓ Greater Emotion Conflict (*EC*) in HCs compared to both clinical groups

**Sex differences:**
✓ Greater *EC* in females than males
✓ Greater *EC* in HCs compared to both clinical groups in females but not males

**Generalizability** and **predictive utility** of computational parameters in the **combined sample**

**Transdiagnostic Generalizability**
✓ Reduced *EC* across specific diagnoses within DEP/ANX and SUDs compared to HCs at follow-up (in some cases only in females)

**Predictive Utility**
✓ Model parameters predictively differentiate DEP/ANX from SUDs
✓ Model parameters predictively differentiate the presence or absence of affective disorders and SUDs, respectively

**Figure 2 Overview of study design and key objectives.** The current study is part of the larger Tulsa 1000 project (Victor et al., 2018). This project collected data from two independent samples, an *exploratory* sample and a *confirmatory* sample (Ns shown in *bottom left* panel), both at a baseline time point and a 1-year follow-up. At each time point, participants completed (among other measures) an Approach Avoidance Conflict (AAC) task and a set of clinical and dimensional scales (*top left* panel). These included the following: the Patient Health Questionnaire-9 (PHQ; depression), the Overall Anxiety Severity and Impairment Scale (OASIS), the Drug Abuse Screening Test (DAST-10), the Anxiety Sensitivity Index (ASI), the Behavioral Activation/Inhibition scales (BIS/BAS), the Patient-Reported Outcome Measurement Information System (PROMIS) depression and anxiety scales, the Positive and Negative Affect Schedule (PANAS), the State-Trait Anxiety Inventory (STAI), and the Temporal Experience of Pleasure Scale (TEPS). For inclusion in the study, the clinical groups were required to meet specific criteria across the PHQ, OASIS, and DAST-10 (*top left*). Formal diagnostic grouping was then performed in accordance with DSM-IV or DSM-5 criteria using the Mini International Neuropsychiatric Interview (MINI), administered at baseline by trained professionals. Building on prior work with this dataset, the pre-registered aims of the current study focused on 1-year follow-up data in the confirmatory sample (*top right* panel). Namely, we tested the replicability of several longitudinal results previously reported on the exploratory dataset. These centered on to two parameters derived from computationally modeling AAC task behavior: decision uncertainty (*DU*) and emotion conflict (*EC*). Following completion of all pre-registered analyses, we then combined the exploratory and confirmatory samples, affording power to test additional questions (*bottom right* panel). Check marks and Xs indicate which of these analyses were and were not successfully replicated, respectively.

# RESULTS

## PARTICIPANT CHARACTERISTICS

Participant demographic and clinical measures are shown in Table 2. Descriptive statistics for model-based measures (*DU* and *EC*) are shown in Table 3. These measures were weakly correlated (*r* = 0.14, *p* = 0.020). As in our previous reports, parameters were log-transformed to minimize skew for subsequent analyses using the *optLog* R package (https://github.com/kforthman/optLog). The resulting log-transformed values were: HCs (*DU* = 0.73 ± 0.96; *EC* = 1.06 ± 0.87), DEP/ANX (*DU* = 0.70 ± 1.03; *EC* = 0.83 ± 0.78), SUDs (*DU* = 0.79 ± 0.99; *EC* = 0.43 ± 0.71).

|  | HCs | DEP/ANX | SUDs | *p*\* |
|---|---|---|---|---|
| Baseline | N = 97 | N = 208 | N = 175 | |
| Follow-up | N = 69 | N = 135 | N = 83 | |
| **Age** | | | | |
| Baseline | 32.09 (11.10) | 33.74 (10.17) | 33.76 (8.42) | 0.33 |
| Follow-up | 34.50 (11.30) | 34.69 (10.17) | 34.41 (8.56) | 0.978 |
| **Sex (Male)+** | | | | |
| Baseline | 38 (39.20) | 51 (24.50) | 68 (38.90) | 0.004\*\* |
| Follow-up | 23 (33.30) | 29 (21.50) | 22.00 (26.50) | 0.184 |
| **PHQ-9** | | | | |
| Baseline | 1.23 (1.82) | 12.53 (5.00) | 6.59 (5.76) | <0.001\*\*\* |
| Follow-up | 0.91 (1.73) | 8.47 (5.62) | 3.30 (4.44) | <0.001\*\*\* |
| **OASIS** | | | | |
| Baseline | 1.10 (1.62) | 9.65 (3.58) | 5.77 (4.46) | <0.001\*\*\* |
| Follow-up | 0.86 (1.75) | 7.11 (4.36) | 3.26 (3.73) | <0.001\*\*\* |
| **DAST-10** | | | | |
| Baseline | 0.19 (0.49) | 0.41 (0.95) | 7.46 (2.23) | <0.001\*\*\* |
| Follow-up | 0.30 (0.55) | 0.37 (0.68) | 2.27 (2.53) | <0.001\*\*\* |
| **WRAT reading score** | | | | |
| Baseline | 62.81 (5.31) | 62.75 (5.06) | 56.79 (6.78) | <0.001\*\*\* |
| Follow-up | 63.25 (5.63) | 63.02 (5.03) | 57.12 (7.12) | <0.001\*\*\* |

**Table 2** Summary statistics and group differences for demographic and clinical measures.

**Note:** For ease of interpretation, commonly used severity cutoffs for each symptom measure are as follows. Patient Health Questionnaire (PHQ): 0–4 indicates minimal depression symptoms, 5–9 indicates mild depression, 10–14 indicates moderate depression, 15–19 indicates moderately severe depression, and scores > 19 indicate severe depression (Kroenke et al., 2001). Overall Anxiety Severity and Impairment scale (OASIS): ≥ 8 is suggestive of clinically significant anxiety (more specific severity ranges are not provided; Campbell-Sills et al., 2009). Drug Abuse Screening Test (DAST): 1–2 indicates low levels of impairment, 3–5 indicates moderate levels of severity, 6–8 indicates substantial levels of severity, and scores of 9–10 indicates severe levels of severity (Skinner, 1982). Please note that not all individuals in the DEP/ANX group displayed elevated symptoms for both anxiety and depression (i.e., had co-morbid anxiety and depression) as measured by OASIS and PHQ, respectively.

WRAT = Wide Range Achievement Test; \**p*-values are based on ANOVAs testing for significant differences between the three groups (\*\*\**p* < 0.001; \*\**p* < 0.01; \**p* < 0.05); +Sex is reported in terms of counts and percentages. Reported *p*-values are based on Chi-Squared tests.

|  | HCs | DEP/ANX | SUDs | *p*\* |
|---|---|---|---|---|
| Baseline | N = 97 | N = 208 | N = 175 |  |
| Follow-up | N = 69 | N = 135 | N = 83 |  |
| **EC** |  |  |  |  |
| Baseline | 3.53 (3.57) | 2.97 (3.06) | 1.69 (1.94) | <0.001\*\*\* |
| Follow-up | 3.75 (3.22) | 2.90 (3.05) | 1.75 (2.10) | <0.001\*\*\* |
| **DU** |  |  |  |  |
| Baseline | 3.96 (4.65) | 4.14 (4.57) | 5.26 (5.72) | 0.048\* |
| Follow-up | 3.22 (3.88) | 3.51 (4.79) | 3.63 (4.67) | 0.855 |
| **Model Accuracy** |  |  |  |  |
| Baseline | 0.77 (0.24) | 0.76 (0.24) | 0.72 (0.24) | 0.185 |
| Follow-up | 0.83 (0.22) | 0.82 (0.24) | 0.80 (0.23) | 0.697 |
| **Action Probability** |  |  |  |  |
| Baseline | 0.64 (0.29) | 0.62 (0.29) | 0.56 (0.27) | 0.019\* |
| Follow-up | 0.70 (0.28) | 0.70 (0.29) | 0.67 (0.27) | 0.696 |

**Table 3** Summary statistics and group differences for model parameters and performance.

**Note:** EC = Emotion Conflict; DU = Decision Uncertainty; *p*-values are based on ANOVAs testing for significant differences between the three groups. \**p* < 0.05, \*\**p* < 0.01, \*\*\**p* < 0.001.

Participant characteristics at baseline for those who did vs. did not return for the follow-up visit are provided in **Supplementary Tables 2–3**. On average, participants who did not return for the follow-up visit had higher scores on the DAST ($t(368.68) = 4.15$, $p < 0.001$, $d = 0.398$) and lower scores on the WRAT, which was used to approximate pre-morbid cognitive capacity ($t(325.23) = -2.11$, $p = 0.036$, $d = -0.216$). There was also a sex difference, with higher attrition in males, both overall ($\chi^2(1, N = 480) = 14.77$, $p < 0.001$) and when restricting to those with SUDs ($\chi^2(1, N = 175) = 9.17$, $p = 0.002$). However, males who did vs. did not return for the follow-up visit did not differ in computational measures or in anxiety, depression, or substance use severity ($|ts| \leq 1.68$, $ps \geq 0.108$). No other demographic or symptom severity differences were found between males and females or between those who did or did not return for follow-up in each clinical group separately ($|ts| \leq 1.59$, $ps \geq 0.118$).

Note that, given this differential attrition, the linear mixed effects analyses (LMEs) detailed below were performed using all participant data, including baseline data from those who did not return for follow-up. This was based on the intention-to-treat principle, with the goal of minimizing potential biasing effects of drop out on longitudinal findings (i.e., as LMEs can incorporate baseline data in model estimation despite partial missing data at follow-up). To examine trajectories of individual change, supplementary analyses were also performed when only including those who returned for the follow-up visit.

## COMPARISON BETWEEN EXPLORATORY AND CONFIRMATORY SAMPLES

Participant characteristics at follow-up in the exploratory and confirmatory samples are compared in Table 4. Individuals in the confirmatory sample tended to be younger ($t(608.34) = 2.05$, $p = 0.041$, $d = 0.165$) and had lower scores on the OASIS ($t(506.89) = 2.20$, $p < 0.028$, $d = 0.192$). When restricting analyses to the individual clinical groups, it was specifically individuals in the DEP/ANX group in the confirmatory dataset who were younger than those in the exploratory dataset ($t(307.16) = 2.05$, $p = 0.041$, $d = 0.225$). No other demographic differences were found (**Supplementary Table 4**).

Figure 3 illustrates the difference in model parameter values between exploratory and confirmatory samples. Here we evaluate sample differences for each group at baseline (all participant data), follow-up, and the change from baseline to follow-up (i.e., only in participants that returned for follow-up). At baseline, lower *DU* values were observed in the confirmatory sample than in the exploratory sample for both DEP/ANX ($t(417.3) = 2.86$, $p = 0.004$, $d = 0.27$, Bayes' Factor [BF] = 5.89)

| RETURNED FOR FOLLOW-UP | EXPLORATORY (N = 324) | CONFIRMATORY (N = 287) | p* |
|---|---|---|---|
| Age | 36.30 (10.92) | 34.57 (9.99) | 0.041* |
| Sex (Male) | 108 (33.30) | 74 (25.80) | 0.051 |
| PHQ-9 | 5.55 (5.96) | 5.12 (5.62) | 0.396 |
| OASIS | 5.36 (4.88) | 4.46 (4.52) | 0.028* |
| DAST-10 | 1.16 (2.03) | 0.91 (1.70) | 0.118 |
| WRAT Reading score | 61.96 (5.43) | 61.41 (6.40) | 0.279 |

**Table 4** Differences in participant characteristics in the exploratory and confirmatory samples at 1-year follow-up.

**Note:** PHQ = Patient Health Questionnaire; OASIS = Overall Anxiety Severity and Impairment Scale; DAST = Drug Abuse Screening Test; WRAT = Wide Range Achievement Test; *p-values are based on a t-test for significant differences between the three groups (*p < 0.05, **p < 0.01, ***p < 0.001). Sex is reported in terms of counts and percentages. Reported p-values are based on Chi-Squared tests.

and SUDs ($t(331.86) = 2.6$, $p = 0.010$, $d = 0.283$, BF = 2.92). At follow-up, HCs also had higher $DU$ in the confirmatory sample when compared to the exploratory sample ($t(99.97) = -2.05$, $p = 0.043$, $d = -0.386$, BF = 1.30). No other sample differences were observed for $DU$ or $EC$ at baseline or follow-up ($|ts| < 1.13$, $ps > 0.26$, BFs < 0.24). However, some further sample differences were identified in how parameter values changed between baseline and follow-up (see Figure 3). First, while $DU$ values decreased from baseline to follow-up in all groups and in both samples, these decreases were stronger in the exploratory sample than in the confirmatory sample for HCs ($F(1, 114) = 6.83$, $p = 0.010$) and DEP/ANX ($F(1, 324) = 3.97$, $p = 0.030$). In contrast, SUDs showed similar decreases in $DU$ over time in both samples.

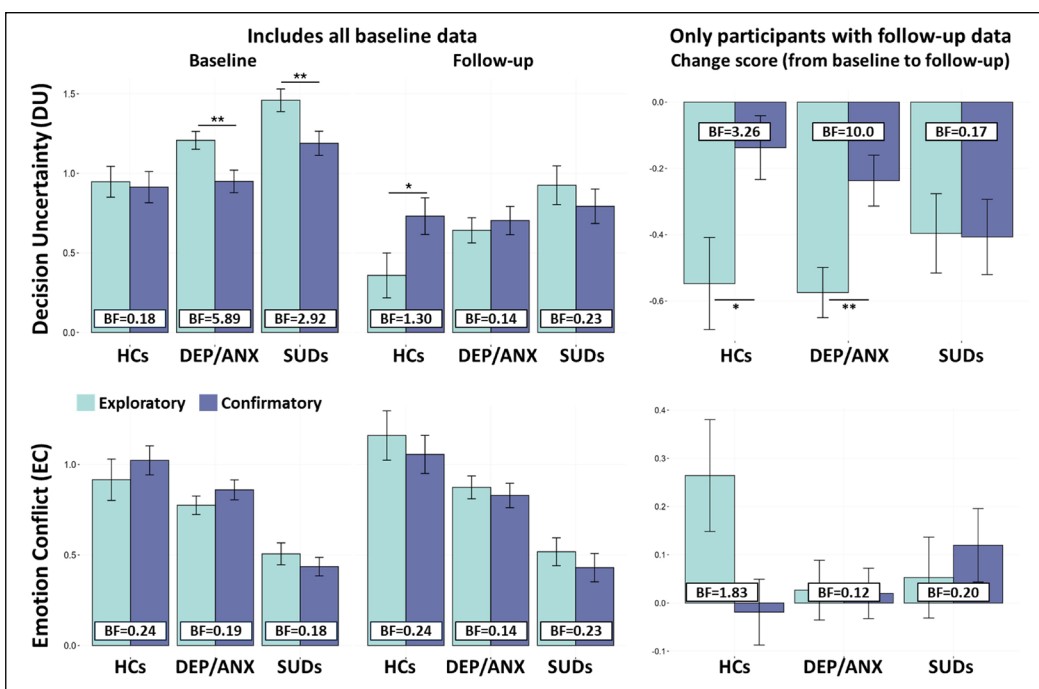

**Figure 3 Comparison between exploratory and confirmatory samples.** *Top*: The left panels compare $DU$ values between groups in each sample at baseline and follow-up. The right panel displays the associated change scores (i.e., follow-up scores minus baseline scores). Here, DEP/ANX and SUDs showed lower $DU$ in the confirmatory sample compared to the exploratory sample at baseline, while HCs showed higher $DU$ in the confirmatory sample compared to the exploratory sample at follow-up. Reductions in $DU$ over time were greater in the exploratory sample than in the confirmatory sample for HCs and DEP/ANX, but not for SUDs. *Bottom*: The left and right panels show equivalent group by sample comparisons for $EC$ scores and change score comparisons, respectively. No sample differences in $EC$ were observed for any groups at baseline or follow-up. HCs in the exploratory sample showed an increase in $EC$ from baseline to follow-up, while participants in the confirmatory sample instead showed no change in $EC$ values. *p < 0.05, **p < 0.01, ***p < 0.001.

## PRE-REGISTERED REPLICATION ANALYSES

### Intra-class correlation analyses between parameters at baseline and 1-year follow-up

Between baseline and 1-year follow-up, ICCs for $EC$ and $DU$ in the confirmatory sample were fair-to-good (ICCs = 0.70 and 0.57, respectively). ICCs for response times (RTs) were fair across task conditions (ICCs between 0.43 and 0.51). For descriptive choice measures, ICCs across conditions were fair-to-good for average chosen runway position (ICCs between 0.54 and 0.71) and poor-to-fair for choice variability (ICCs between 0.37 and 0.51). These results were consistent with those in the exploratory sample. Consistency in clinical scale scores and dimensional measures was also fair-to-good across all participants (ICCs between 0.42 and 0.74). Detailed results of ICCs by group and condition for task measures, as well as for clinical scales and post-task surveys, are provided in **Supplementary Tables 5–6**.

## Model validation

Across all participants, the model accurately predicted behavior in 81.35% (SD = 1.4%) of trials (i.e., chance accuracy = 11%; 1/9 runway positions) with an average action probability of 0.69 ± 0.28. There were no group differences in model performance (Table 3). As expected, higher values for each parameter predicted slower RTs for each task condition ($DU$: $r$s = 0.31–0.55, $p$s < 0.001; $EC$: $r$s = |0.14–0.30|, $p$s < 0.024); see **Supplementary Figure 2**. This is consistent with results at baseline and with prior longitudinal results in the exploratory sample (Smith et al., 2021b; Smith et al., 2021c; Smith et al., 2023). Relationships between model parameters and post-task survey questions were also in expected directions, as detailed in Table 5. In particular, there were significant positive associations between $EC$ and both self-reported anxiety (Q2) and avoidance motivation (Q5) during the task, and a positive association between $DU$ and greater self-reported decision difficulty (Q3) on the task (**Supplementary Figure 3**). As the model was not fit to these RTs and self-report measures, this provides external support for its validity. Supplementary within-group analyses for these measures are reported in **Supplementary Table 7**.

Mehta et al.
*Computational Psychiatry*

172

| POST-TASK SELF-REPORT QUESTIONS (LIKERT SCALE: 1 = NOT AT ALL; 7 = VERY MUCH) | MEAN (SD) BASELINE | | MEAN (SD) 1-YEAR FOLLOW-UP (N = 287) | EMOTION CONFLICT PARAMETER (*EC*) | DECISION UNCERTAINTY PARAMETER (*DU*) |
|---|---|---|---|---|---|
| | ALL PARTICIPANTS (N = 480) | RETURNED FOR FOLLOW-UP (N = 287) | | | |
| Q1. I found the positive pictures enjoyable | 5.02 (1.56) | 5.17 (1.56) | 4.97 (1.59) | 0.06 | –0.04 |
| Q2. The negative pictures made me feel anxious or uncomfortable | 4.01 (1.97) | 3.98 (1.99) | 4.00 (1.95) | **0.34\*\*\*** | **0.06** |
| Q3. I often found it difficult to decide which outcome I wanted | 2.32 (1.71) | 2.26 (1.68) | 2.07 (1.60) | 0.04 | **0.42\*\*\*** |
| Q4. I always tried to move all the way towards the outcome with the largest reward points | 4.86 (2.35) | 4.84 (2.38)⁺ | 4.98 (2.44) | **–0.77\*\*\*** | **–0.46\*\*\*** |
| Q5. I always tried to move all the way away from the outcome with the negative picture/sounds | 2.86 (2.15) | 2.92 (2.19) | 2.93 (2.32) | **0.72\*\*\*** | **0.31\*\*\*** |
| Q6. When a negative picture and sound were displayed, I kept my eyes open and looked at the picture | 5.43 (1.89) | 5.39 (1.89)⁺ | 5.29 (1.97) | **–0.42\*\*\*** | **–0.21\*\*\*** |
| Q7. When a negative picture and sound were displayed, I tried to think about something unrelated to the picture to distract myself | 2.84 (1.90) | 2.93 (1.95) | 2.95 (1.97) | **0.32\*\*\*** | 0.04 |
| Q8. When a negative picture and sound were displayed, I tried other strategies to manage emotions triggered by the pictures | 3.04 (1.91) | 3.11 (1.90) | 3.22 (1.93) | **0.34\*\*\*** | **0.05** |

**Table 5** Post-task self-report questionnaire items at baseline and follow-up, and correlations with computational model parameters at follow-up.

**Note:** ⁺$p$ < 0.05, ⁺⁺$p$ < 0.01, ⁺⁺⁺$p$ < 0.001 (pre-post differences); \*$p$ < 0.05, \*\*$p$ < 0.01 \*\*\*$p$ < 0.001 (correlations at follow-up). Results that were statistically significant in the exploratory sample are highlighted in bold.

## Replication of longitudinal stability and interactions with sex

To evaluate stability in model parameters, LMEs predicting each parameter were estimated including main effects of age, sex, group, and time, as well as interactions between group and sex and group and age (see Table 6). Most notably, the LME predicting $DU$ showed significant main

effects of time ($F$(1, 346) = 32.83, $p < 0.001$) and group ($F$(2, 469) = 3.08, $p = 0.047$). This indicated a reduction in *DU* over time ($EMM_{Baseline}$ = 1.03; $EMM_{Follow-up}$ = 0.73), and greater *DU* in SUDs ($EMM$ = 1.05) than in the DEP/ANX group ($EMM$ = 0.84).

Mehta et al. **173**
*Computational Psychiatry*

| PREDICTOR* | TEST, *P* VALUE | EMM | POST-HOC CONTRASTS |
|---|---|---|---|
| **Decision Uncertainty (*DU*)** | | | |
| Group | $F$(2, 469) = 3.08, $p = 0.047$ | D/A = 0.84; HCs = 0.88; SUDs = 1.05 | D/A – HCs: $t$(441.22) = –0.387, $p = 0.699$, $d = -0.07$<br>D/A – SUDs: $t$(472.34) = -2.23, $p = 0.026$, $d = -0.33$<br>HCs – SUDs: $t$(453.13) = –1.441, $p = 0.150$, $d = -0.26$ |
| Time | $F$(1, 346) = 32.83, $p < 0.001$ | T1 = 1.03; T2 = 0.73 | T1 – T2: $t$(346.52) = 5.73, $p < 0.001$, $d = 0.46$ |
| Age | $F$(1, 462) = 28.76, $p < 0.001$ | | |
| Sex | $F$(1, 470) = 3.23, $p = 0.073$ | | NS |
| Group × Age | $F$(2, 467) = 2.84, $p = 0.060$ | | NS |
| Group × Sex | $F$(2, 471) = 0.14, $p = 0.873$ | | NS |
| **Emotion Conflict (*EC*)** | | | |
| Group | $F$(2, 470) = 15.86, $p < 0.001$ | D/A = 0.85; HCs = 1.07; SUDs = 0.44 | D/A – HCs: $t$(448.69) = -2.468, $p = 0.014$, d = –0.49<br>D/A – SUDs: $t$(473.21) = 5.44, $p < 0.001$, d = 0.92<br>HCs – SUDs: $t$(458) = 6.793, $p < 0.001$, d = 1.41 |
| Time | $F$(1, 333) = 0.16, $p = 0.688$ | | NS |
| Age | $F$(1, 465) = 5.28, $p = 0.022$ | | |
| Sex | $F$(1, 471) = 6.24, $p = 0.013$ | Female = 0.80; Male = 0.66 | Female – Male: $t$(475.87) = 2.03, $p = 0.043$, $d = 0.34$ |
| Group × Age | $F$(2, 473) = 1.64, $p = 0.194$ | | NS |
| Group × Sex | $F$(2, 472) = 5.41, $p = 0.005$ | <u>Female:</u><br>D/A = 0.90; HCs = 1.21; SUDs = 0.41<br><u>Male:</u><br>D/A = 0.72; HCs = 0.72; SUDs = 0.53 | <u>Female:</u><br>D/A – HCs: $t$(443.07) = –2.891, $p = 0.004$, $d = -0.71$<br>D/A – SUDs: $t$(464.91) = 5.515, $p < 0.001$, $d = 1.13$<br>HCs – SUDs: $t$(450.94) = 7.009, $p < 0.001$, $d = 1.83$<br><u>Male:</u><br>D/A – HCs: $t$(467.08) = –0.018, $p = 0.986$, $d = -0.01$<br>D/A – SUDs: $t$(492.88) = 1.403, $p = 0.161$, $d = 0.43$<br>HCs – SUDs: $t$(483.56) = 1.311, $p = 0.191$, $d = 0.44$ |

**Table 6** Results of linear mixed effects models predicting DU and EC in data including all baseline participants, when accounting for effects of group, time, age, and sex.

**Note:** HCs = Healthy Comparisons; D/A = Depression and Anxiety; SUDs = Substance Use Disorders; T1 = Baseline; T2 = Follow-up; NS = non-significant.

For interpretability, age and WRAT scores were centered; sum coding was used for sex (female = −1; male = 1), time (baseline = −1; follow-up = 1), and group (with HCs coded as –1).

In the LME predicting *EC*, there was a main effect of group ($F$(2, 470) = 15.86, $p < 0.001$), reflecting greater values in HCs ($EMM$ = 1.07) than in both clinical groups ($EMM_{DEP/ANX}$ = 0.85; $EMM_{SUDs}$ = 0.44). Further, SUDs had significantly lower *EC* than DEP/ANX. There was a main effect of sex ($F$(1, 471) = 6.24, $p = 0.013$), whereby females displayed greater *EC* than males ($EMM_{Female}$ = 0.80; $EMM_{Male}$ = 0.66). A *group × sex* interaction ($F$(2, 472) = 5.41, $p = 0.005$) was also present. This interaction indicated that the group effect was primarily driven by females (Females: $EMM_{HCs}$ = 1.21; $EMM_{DEP/ANX}$ = 0.90; $EMM_{SUDs}$ = 0.41; and Males: $EMM_{HCs}$ = 0.72; $EMM_{DEP/ANX}$ = 0.72; $EMM_{SUDs}$ = 0.53; see Figure 4).

Together, these results replicated the following from previous work: (1) reduction in *DU* over time; (2) greater *DU* in SUDs than DEP/ANX (although differences with HCs did not replicate); (3) lower *EC* in both clinical groups than HCs (lowest in SUDs); (4) lower *EC* in males than females; (5) group differences in *EC* were driven by females. A further replication summary is provided in **Supplementary Figure 4**.

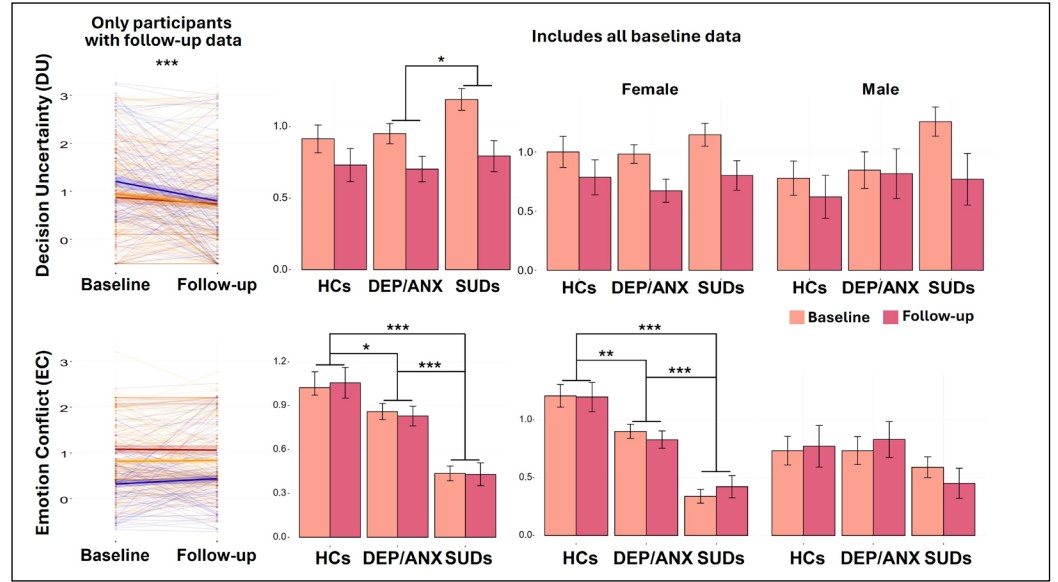

**Figure 4 Pre-registered tests of group, time, and sex effects on model parameters.** Spaghetti plots (left) show changes from baseline to follow-up and therefore only include participants who returned for the follow-up visit. Bar plots (right) include all baseline participants, including those who did not return for follow-up. These plots illustrate individual differences in the stability of parameter values over time. Here, thick lines within the spaghetti plots indicate group means and lighter lines indicate individual values. Shaded areas around the line for each group mean reflect the associated standard errors. Sex comparisons in the lower bar graphs illustrate how observed group differences in *EC* were mainly driven by females. *$p < 0.05$, **$p < 0.01$, ***$p < 0.001$.

Similar results were also found in pre-registered LMEs including only effects of group and time, and in further LMEs controlling for possible effects of premorbid cognitive ability (WRAT Reading scores) in those with available data (**Supplementary Tables 8–9**). Analogous LMEs were also performed when only including data from participants who returned for the follow-up visit, again with similar results (**Supplementary Figure 5**).

Identical analyses were also performed for the descriptive task measures: RTs, average chosen runway position, and choice variability (**Supplementary Tables 10–21; Supplementary Figure 6**). These results also replicated many of our previously reported findings (detailed in **Supplementary Figure 7**). Briefly, we observed lower choice variability and faster RTs over time across all groups (main effect of time), as well as reduced avoidance behavior in SUDs (reflected in average chosen runway position for AV, CONF2, CONF4, and CONF6 trials), which replicated prior results. Consistent with results for *EC* above, a *group × sex* interaction was present in the CONF2 and CONF4 conditions, wherein reduced avoidance behavior (as measured by average chosen runway position) in SUDs was observed in female participants only.

In contrast, when testing the ability of baseline model parameters or other behavioral task measures to predict change in symptoms over time, most previously reported findings did not replicate (**Supplementary Figure 1**). The only replicable finding was a negative relationship between average chosen runway position in the CONF2 condition at baseline and change in self-reported BIS scores at follow-up ($r = -0.14$; $p = 0.047$). Other significant associations were observed (i.e., present in the confirmatory sample but not in the exploratory sample), however, suggesting the presence of either false positives or false negatives in either sample. Lastly, LMEs were also performed for post-task self-report survey items assessing anxiety (Q2), decision-making difficulty (Q3), approach motivation (Q4), and avoidance motivation (Q5; **Supplementary Tables 22–24**). Results replicated previous findings in some cases, but not others (for details, see **Supplementary Figure 8**). Successful replications included main effects of group on approach ($F(2, 468) = 4.87$, $p = 0.008$) and avoidance ($F(2, 467) = 4.18$, $p = 0.016$) motivation, where SUDs (EMM$_{Approach}$ = 5.44; EMM$_{Avoid}$ = 2.39) displayed greater approach motivation (EMM$_{SUDs}$ = 5.44; EMM$_{HCs}$ = 4.32; $t(452.58) = -4.03$, $p < 0.001$, $d = -0.73$) and reduced avoidance motivation (EMM$_{SUDs}$ = 2.34; EMM$_{HCs}$ = 3.31; $t(448.86) = 3.81$, $p < 0.001$, $d = 0.63$) compared to HCs. The previously observed *group × sex* interaction ($F(2, 469) = 4.32$, $p = 0.014$) in avoidance motivation was also observed here, suggesting that the lower avoidance motivation observed in SUDs (EMM$_{Female}$ = 2.20; EMM$_{Male}$ = 2.67) than HCs (EMM$_{Female}$ = 3.64; EMM$_{Male}$ = 2.56) was driven by females (Females: $t(438.38) = 4.49$, $p < 0.001$, $d = 0.93$; Males: $t(486.53) = -0.254$, $p = 0.799$, $d = -0.07$).

In summary, similar to our results above, reduced avoidance motivation within SUDs, specifically within female participants, was consistent when measured using the post-task self-report questionnaire. Together, these results therefore robustly support the previously observed group and sex differences in *EC*. They also motivate further investigation into the potential clinical relevance of these findings. Having performed each of these pre-registered replication analyses, we next combined the exploratory and confirmatory samples to take advantage of the added power this provided to answer additional questions, which focused on testing the transdiagnostic generalizability and predictive utility of our computational measures.

## COMBINED SAMPLE ANALYSES

### Transdiagnostic generalizability of model parameters in the combined sample

After carrying out all pre-registered analyses above, we then combined exploratory ($N_{Baseline}$ = 478; $N_{Follow-up}$ = 324) and confirmatory ($N_{Baseline}$ = 480; $N_{Follow-up}$ = 287) samples. This provided greater power to examine whether observed group differences were consistent across the more specific diagnostic categories present within the dataset (e.g., generalized anxiety disorder, social anxiety disorder, methamphetamine use disorder, opioid use disorder, etc.). To answer this question, separate LMEs were performed including only HCs and different subsets of participants from the combined clinical groups. Namely, one LME was performed including the subset of participants who met criteria for each specific disorder (e.g., all those with stimulant [cocaine or methamphetamine] use disorder, or all those with major depressive disorder, etc.). Note that, due to comorbidities, these subsets of participants had partial overlap (e.g., individuals with major depressive disorder may or may not have had other anxiety or substance use disorders). These LMEs then predicted each model parameter based on the specific diagnostic category in question (e.g., HCs vs. generalized anxiety, HCs vs. opioid use disorder, etc.); effects of time and the associated interaction term were also included. Subsequent post-hoc power analyses also considered attrition rate within each sample when testing for effects of group (at a 0.05 significance level) on *DU* and *EC*, respectively. As a secondary check, these LMEs were also re-performed when including possible effects of age, sex, and their interaction with group. Here we found consistent main effects of group across specific disorders for each parameter (Figures 5 and 6).

There were no effects of time, with the exception of increases in *EC* from baseline to follow-up in those with sedative use disorders ($EMM_{Baseline}$ = 0.78; $EMM_{Follow-up}$ = 0.90) and major depression ($EMM_{Baseline}$ = 0.76; $EMM_{Follow-up}$ = 0.83). Overall, these results indicated that HCs had greater *EC* values than each specific diagnostic group ($Fs \geq 8.03$, $ps \leq 0.002$). Post-hoc power analyses indicated that the associated sample sizes afforded power of $\beta > 0.977$ to detect these group differences, with the exception of hallucinogen use disorder ($\beta = 0.686$). When accounting for effects of age and sex, these main effects of group remained in all cases ($Fs \geq 6.41$, $ps \leq 0.012$). Females also displayed greater avoidance than males across all specific diagnostic groups ($Fs \geq 17.41$, $ps < 0.001$). Extending prior results, a significant *group × sex* interaction ($Fs \geq 5.16$, $ps < 0.024$) was also found in most cases. Specifically, the group differences between HCs and those with major depression, generalized anxiety, panic, cannabis, stimulant, and opioid use disorders were each driven by females ($ts \geq 4.22$, $ps < 0.001$). In alcohol ($F(1, 270) = 9.57$, $p = 0.002$) and sedative ($F(1, 244) = 8.20$, $p = 0.005$) use disorder, the group difference was present in both sexes, but the effect in females remained numerically stronger (alcohol: $t(268.68) = 6.38$, $p < 0.001$, $d = 1.81$; sedative: $t(237.21) = 6.97$, $p < 0.001$, $d = 1.89$) than in males (alcohol: $t(274.97) = 2.37$, $p = 0.018$, $d = 0.62$; sedative: $t(250.48) = 2.18$, $p = 0.03$, $d = 0.70$).

Group differences were also found in *DU*, whereby HCs had lower *DU* than most sub-groups ($Fs \geq 4.08$, $ps \leq 0.044$). A similar but non-significant pattern was found in panic disorder ($F(1, 271) = 3.47$, $p = 0.064$). Generalized and social anxiety disorders did not show significant effects ($Fs \geq 0.14$, $ps \leq 0.712$). Post-hoc simulations suggested power of $\beta > 0.846$ to detect group differences with the observed effect sizes, with the exception of generalized and social anxiety disorders, which had associated power levels of only 0.584 and 0.086 (i.e., due to the small observed effect sizes). A main effect of time ($Fs \geq 19.80$, $ps < 0.001$) was also found, with lower *DU* at follow-up in all cases.

Mehta et al. **176**
*Computational Psychiatry*

Comparable results were found when accounting for effects of age, sex, and their interaction with group. There was also a *group × age* interaction ($Fs \geq 6.52$, $ps \leq 0.011$) in all groups with affective disorders, with the exception of individuals with PTSD ($F(1, 274) = 3.58$, $p = 0.059$). This interaction reflected a positive association between age and *DU* in affective disorders (depression, generalized anxiety, social anxiety, and panic disorders; $|ts| \geq 2.55$, $ps \leq 0.011$) but not in HCs. No other significant effects were found. Notably, our previous paper reported that *DU* (but not *EC*) differentiated social anxiety from other affective disorders at baseline within a logistic regression model (Smith et al., 2023). As a supplementary test, we repeated this analysis in the follow-up data here to confirm whether this difference was stable over time (i.e., including the effects of *DU* and *EC*, along with their interaction with age and sex). This revealed a significant effect of age (Wald $z = -3.7$, $p < 0.001$), but no other main effects or interactions were present (Wald $z = -1.68$ to 1.60, $ps > 0.093$).

Descriptive statistics for model parameters in each specific diagnostic group are provided in **Supplementary Table 25**. Full results of all LMEs are provided in **Supplementary Tables 26–29.** Results of associated post-hoc power analyses are provided in **Supplementary Table 30**.

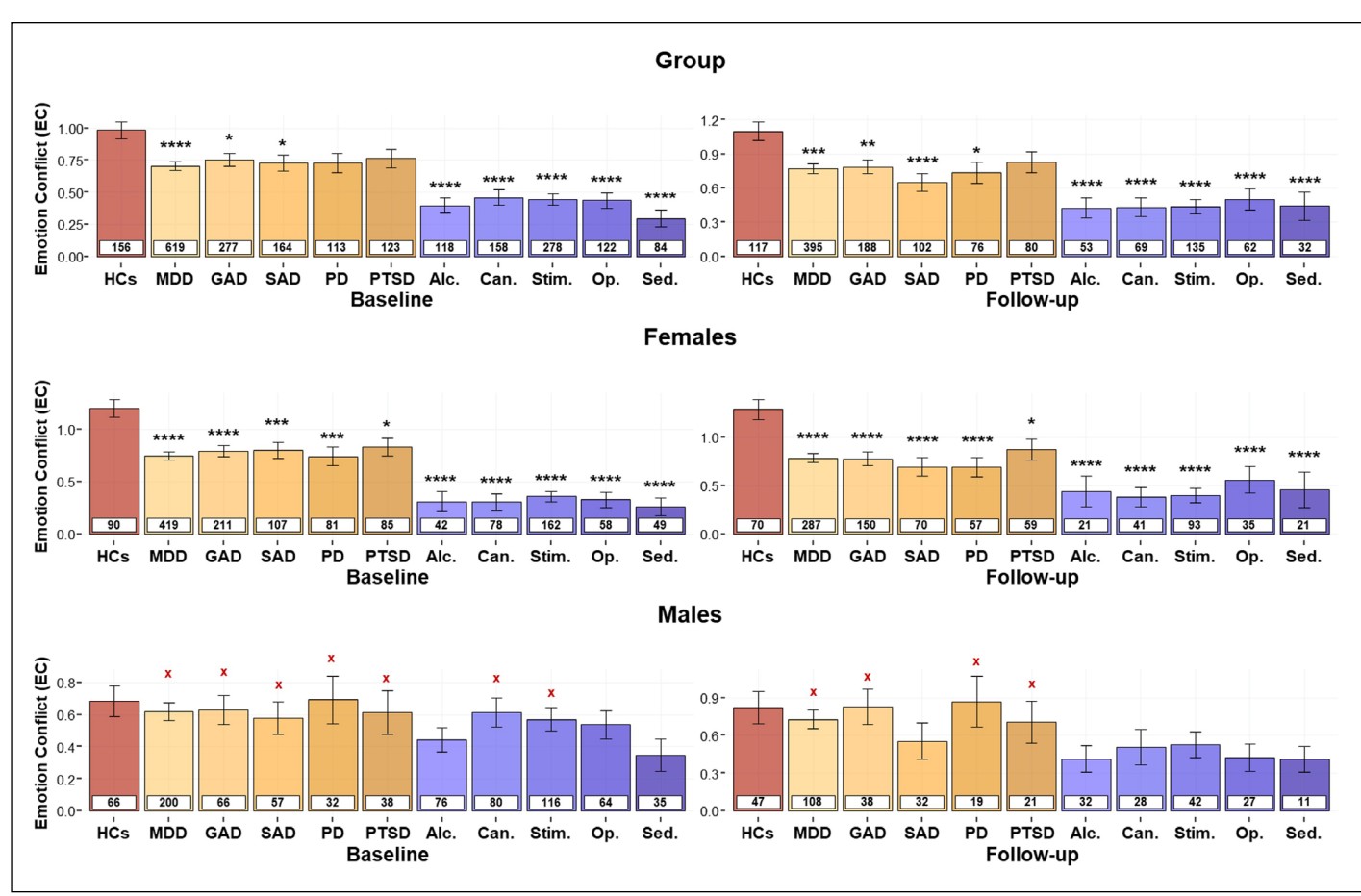

**Figure 5 Effects of group and sex on emotion conflict (*EC*) across specific diagnostic categories in the combined sample.** Values are shown separately for healthy comparisons (HCs) and each specific affective and substance use disorder. For each group, the number of participants is provided in the white box. ***Top***: Differences between HCs and each specific diagnosis are consistent with differences seen in the broader clinical groupings. ***Middle***: The overall group differences appear to be driven by females at both baseline and follow-up. ***Bottom***: Males with affective and substance use disorders did not differ from HCs. MDD: Major Depressive Disorder; GAD: Generalized Anxiety Disorder; SAD: Social Anxiety Disorder; PD: Panic Disorder; PTSD: Post-Traumatic Stress Disorder; Alc.: Alcohol; Can.: Cannabis; Stim.: Stimulant; Op.: Opioid; Sed.: Sedative. For illustrative purposes, Bayesian *t*-tests were performed to evaluate evidence for the presence or absence of group differences with HCs at each time point. Bayes Factors (BFs) greater than 3 were taken to support the presence of an effect, denoted by black stars (*): *BF > 3; **BF > 10, ***BF > 30, ****BF > 100. BFs between 0.33 and 3 were considered equivocal null results. BFs less than 0.33 were taken to support the absence of a group difference with HCs, denoted by red Xs: ˣBF < 0.33, ˣˣBF < 0.1, ˣˣˣBF < 0.033, ˣˣˣˣBF < 0.001.

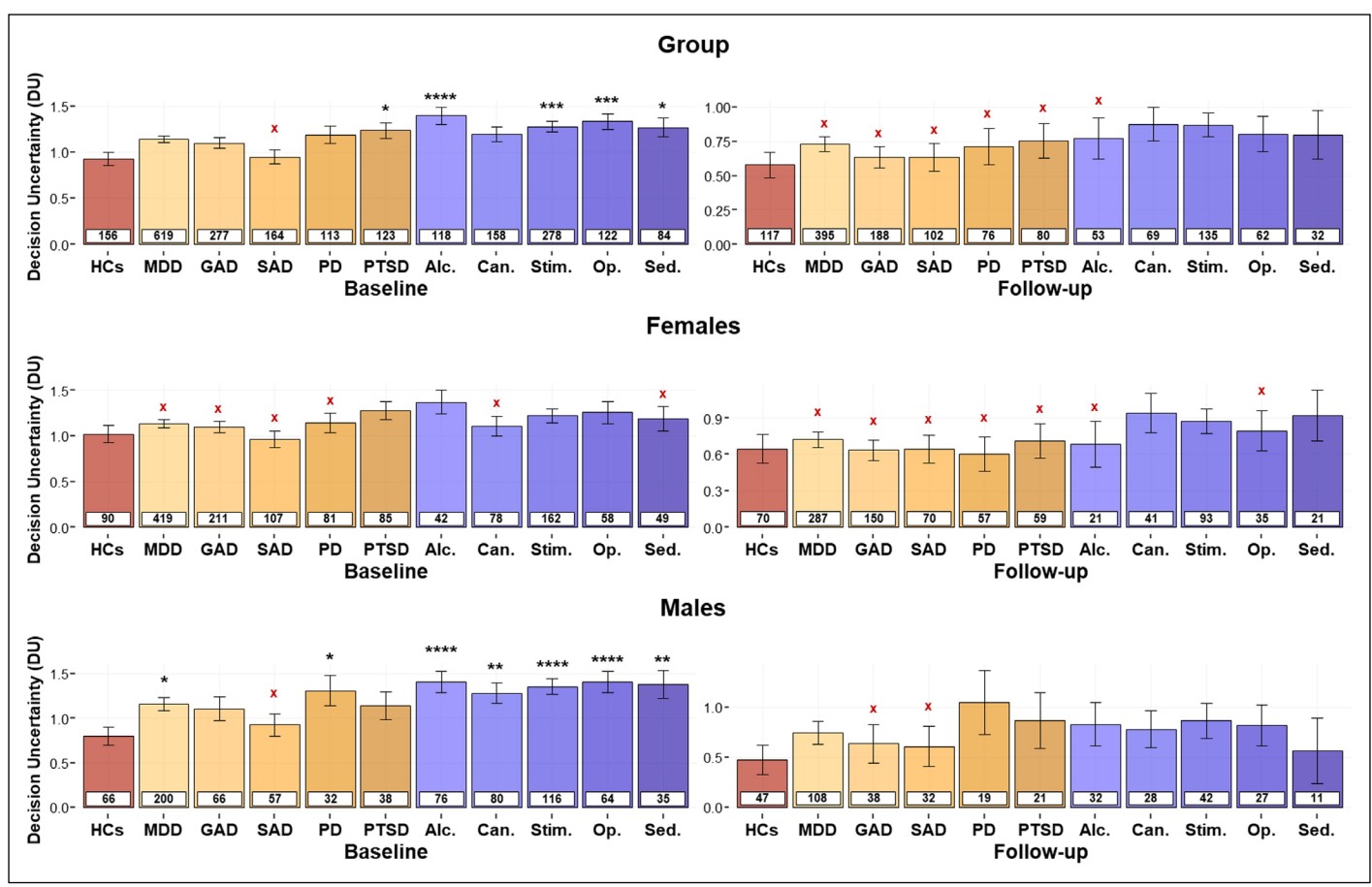

**Figure 6 Effects of group and sex on decision uncertainty (*DU*) across specific diagnostic categories in the combined sample.** Values are shown separately for healthy comparisons (HCs) and each specific affective and substance use disorder. For each group, the number of participants is provided in the white box. **Top**: Group differences (between HCs and each specific diagnosis) were consistent with differences seen in DEP/ANX and SUDs more broadly. **Middle** and **Bottom**: Male but not female SUDs displayed greater *DU* at baseline. MDD: Major Depressive Disorder; GAD: Generalized Anxiety Disorder; SAD: Social Anxiety Disorder; PD: Panic Disorder; PTSD: Post-Traumatic Stress Disorder; Alc.: Alcohol; Can.: Cannabis; Stim.: Stimulant; Op.: Opioid; Sed.: Sedative. For illustrative purposes, Bayesian *t*-tests were performed to evaluate evidence for the presence or absence of group differences with HCs at each time point. Bayes Factors (BFs) greater than 3 were taken to support the presence of an effect, denoted by black stars (*): *BF > 3; **BF > 10; ***BF > 30, ****BF > 100. BFs between 0.33 and 3 were considered equivocal null results. BFs less than 0.33 were taken to support the absence of a group difference with HCs, denoted by red Xs: ˣBF < 0.33, ˣˣBF < 0.1, ˣˣˣBF < 0.033, ˣˣˣˣBF < 0.001.

## Model-based trial-by-trial choice uncertainty predicts response times in the combined sample

As expected, trial-by-trial *choice uncertainty* predicted longer RTs across participants (Intercept: $F(1, 916) = 441.48$, $p < 0.001$; **Supplementary Figure 9**). Notably, this association was significantly higher in SUDs (EMM = 0.16) than in both DEP/ANX (EMM = 0.11; $t(943.433) = -4.67$, $p < 0.001$, $d = -0.32$) and HCs (EMM = 0.11; $t(898.45) = -4.12$, $p < 0.001$, $d = -0.38$).

These results prompted further post-hoc analyses to test for consistent group differences in the relationship between RTs and both choice behavior (average chosen runway position; **Supplementary Figure 10**) and model parameters. Here, LMEs showed a significant *group × RT* interaction in predicting choice of runway position ($F(2, 1547.32) = 4.78$, $p = 0.009$; $ET_{HCs} = -0.59$; $ET_{DEP/ANX} = -1.34$; $ET_{SUDs} = -1.94$) and *EC* ($F(2, 1554.12) = 3.48$, $p = 0.031$; $ET_{HCs} = 0.17$; $ET_{DEP/ANX} = 0.46$; $ET_{SUDs} = 0.69$), but not *DU* ($F(2, 1462.84) = 0.29$, $p = 0.752$). Both significant interactions suggested that shorter RTs were associated with greater approach behavior to a greater extent in SUDs than HCs (choice: $t(1533) = 3.13$, $p = 0.005$; *EC*: $t(1556) = -2.64$, $p = 0.023$).

## Predictive utility of model parameters in differentiating diagnoses within the combined sample

Our predictive (out-of-sample) categorization analyses using the stacked machine learning approach (Kuhn, 2008) first restricted the sample to individuals with one or more clinical diagnoses (i.e., removing HCs) and then classified individuals (Figure 7) in terms of those with affective

Mehta et al.
*Computational Psychiatry*

disorders but no SUDs vs. SUDs but no affective disorders. Here we found that ENET showed the highest balanced accuracy (0.71; Sensitivity = 0.76; Specificity = 0.67) when model parameters (*EC* and *DU* at baseline and follow-up), age, and sex were used as predictors. In comparison, when only age and sex were used as predictors, ENET showed a lower balanced accuracy of 0.47, comparable to a neutral threshold of 0.5 (Sensitivity = 0.30; Specificity = 0.65). Thus, adding model-based information led to improved out-of-sample classification. When further evaluating AUCs as a means of quantifying the ability of the model to distinguish between groups, we found an acceptable discrimination ability of 0.77, compared to poor discrimination with age and sex alone (AUC = 0.50). In addition, the exclusion of either *DU* (balanced accuracy = 0.59; AUC = 0.65) or *EC* (balanced accuracy = 0.51; AUC = 0.52) from the best models each caused a substantial reduction in model performance. Performance metrics for all predictive models are provided in **Supplementary Table 30**.

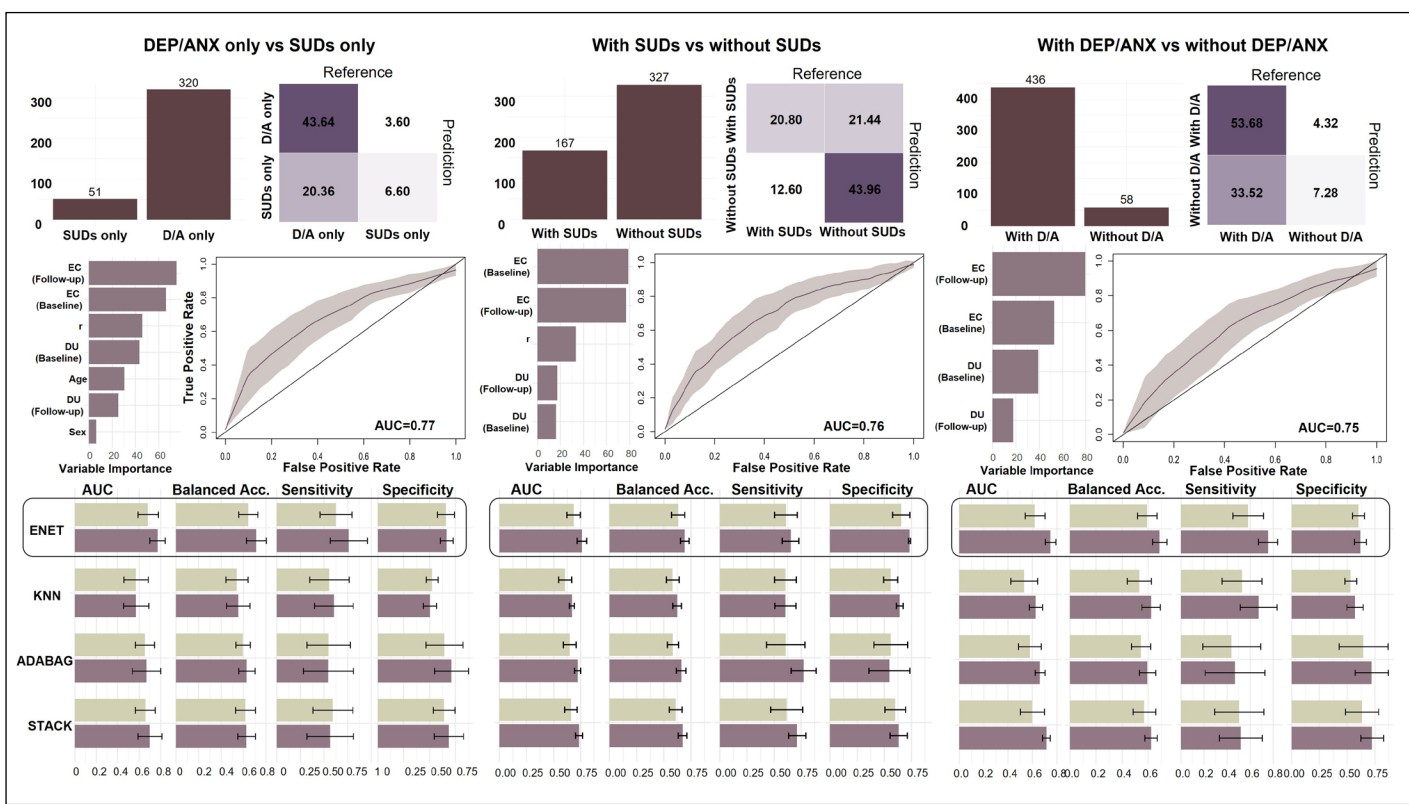

**Figure 7 Predictive classification of diagnostic status in the combined clinical sample.** In the combined clinical sample (excluding HCs), multiple stacked classification algorithms predictively classified (*left panel*) individuals with only affective disorders (DEP/ANX) vs. only substance use disorder (SUDs), (*middle panel*) individuals with vs. without SUDs, and (*right panel*) individuals with vs. without DEP/ANX. ***Top-left***: Histograms show the number of participants in each group. Up-sampling was used to minimize model biases. ***Top-right***: Confusion matrices illustrate classification accuracy for the winning model. ***Middle-left***: Bar plots indicate variable importance (VI), which is the relative contribution each predictor made to the winning model. This metric was computed using the STACK approach to reflect the contribution of predictors across algorithms. ***Middle-right***: Receiver operating characteristic (ROCs) curves are shown for the winning model. ROC curves provide information about true positive and false positive rates at various categorization thresholds. Associated area-under-the-curve (AUC) values show the probability of a sample being assigned to the correct group. ***Bottom***: Model performance was evaluated using AUCs, balanced accuracy, sensitivity (true positive rate), and specificity (true negative rate). Balanced accuracy was used due to unbalanced sample sizes. Performance metrics were averaged across the five folds of cross validation. ENET had the highest balanced accuracy (highlighted) for each predictive classification and was chosen as the best performing algorithm. Algorithms: ENET = elastic net; KNN = k-nearest neighbors; ADABAG = Bagged AdaBoost.

The same approach was then repeated (still excluding HCs) to classify: 1) the presence vs. absence of SUDs (**Supplementary Table 31**), 2) the presence vs. absence of affective disorders (**Supplementary Table 33**), and 3) comorbid affective and substance use disorders vs. the presence of either disorder alone (**Supplementary Table 33**). In each case, ENET similarly showed the best performance, where incorporating model-based information led to improvement from chance (with age and sex alone) to above-chance balanced accuracy (0.69, 0.69, and 0.64, respectively) and discrimination ability (AUCs: 0.76, 0.75, and 0.68, respectively). Exclusion of *DU*

from the best models caused a reduction in balanced accuracy (0.61, 0.58, and 0.60, respectively) and discrimination ability (AUC: 0.64, 0.64, and 0.63, respectively). Similar reductions in balanced accuracy (0.60, 0.59, and 0.63, respectively) and discrimination ability (AUC: 0.63, 0.60, and 0.66, respectively) were found when *EC* was instead excluded from the best models.

When examining variable importance across the three groupings described above, *EC* at baseline (VI ≥ 59.97) and follow-up (VI ≥ 34.69) had the highest relative contribution in all cases when compared to *DU* at baseline (16.21 ≥ VI) or follow-up (29.30 ≥ VI).

## DISCUSSION

The current study was designed to achieve three main goals. First, as pre-registered, we repeated a set of longitudinal analyses in a new sample of individuals with depression and/or an anxiety disorder (DEP/ANX), substance use disorders (SUDs), and healthy comparisons (HCs) who each completed an approach-avoidance conflict (AAC) task at two time points, one year apart. This allowed us to evaluate the reliability of computational task measures and the stability of group differences over time that were previously observed. These analyses successfully replicated some, but not all, hypothesized results.

With respect to reliability, ICCs showed fair-to-good levels of consistency in both model parameters over a 1-year period, similar to the fair levels of consistency found in our exploratory study (Smith et al., 2021c). To our knowledge, no prior computational modeling studies have tested parameter reliability in other tasks over a comparable time period; but these results appear to fall near the middle of a wide range of test-retest reliability levels (from poor to excellent) found for other computational task measures over significantly shorter time periods (Brown et al., 2020; Chung et al., 2017; Enkavi et al., 2019; Moutoussis et al., 2019; Price et al., 2019; Shahar et al., 2019). These results also support our previous examination of test-retest reliability for *EC* and *DU* over a shorter 2–3 week period (Smith et al., 2021c), which demonstrated superior test-retest reliability for the former (ICC = 0.84) and similar reliability of the latter (ICC = 0.54) relative to the longer time period we tested here. With respect to the stability of group differences, results appeared to replicate for one parameter (*EC*) but not for the other (*DU*). We discuss these results in detail below.

For our second goal, we combined participant data from the previous and current samples to answer more granular questions about how computational mechanisms might differentiate specific clinical disorders. Namely, we tested whether individuals with each specific affective or substance use disorder would show the same pattern of group differences relative to HCs as seen in the broader groupings. As discussed below, these results supported a largely transdiagnostic pattern. Finally, for our third goal, we carried out a machine learning-based (out-of-sample) classification approach to evaluate whether computational modeling measures could predictively identify individuals as having affective or substance use disorders (or comorbidity of the two), in which results showed moderate levels of accuracy. We now discuss the results of each set of analyses in more detail.

As briefly summarized above, pre-registered replication analyses confirmed that some, but not all, prior findings were present in the new sample. Most centrally, the greater *EC* values previously seen in HCs compared to both clinical groups showed longitudinal stability, most starkly in females and in those with SUDs. In contrast, *DU* tended to reduce over time, and group estimates converged to non-significant differences at 1-year follow-up. There was also a general pattern wherein specific diagnoses tended to differ from HCs in the same direction as the broader DEP/ANX and SUD groupings, suggesting transdiagnostic relevance. At the same time, model-based measures could also predictively classify whether individuals had DEP/ANX or an SUD, suggesting that some computational decision mechanisms may also differ between diagnoses. We discuss each of these findings below.

In our initial exploratory studies, the pattern of group differences in *EC* was surprising, as maladaptive avoidance is often observed clinically in anxiety and depression. However, given the confirmation of this result, both over time and in independent samples, it appears that higher

*EC* in this task may instead reflect adaptive responding, particularly in females. One possibility is that this could reflect a healthy regulation strategy in HCs, in which anticipated emotional states are more strongly prioritized over small task rewards. If so, results might suggest that those with SUDs, and to a weaker extent DEP/ANX, may not prioritize their emotional states in this same way. For example, individuals in the clinical groups might instead be driven by an aversion to the thought of missing out on reward. A related factor could be that individuals in the clinical groups have less awareness or trust in their bodily/emotional signals (e.g., as suggested by Lavalley et al., 2024; Smith et al., 2021a; Smith et al., 2020), and/or that their elevated baseline negative affect attenuates the impact of unpleasant task stimuli during choice. Yet other possibilities include a potentially greater influence of demand characteristics in the clinical groups (i.e., in which they are more driven to seek points because they believe that is what is expected of them by experimenters); or that lower *EC* stems from blunted responses to negative stimuli, as suggested in some previous work in SUDs (Weinberg et al., 2016). This would be congruent with the broader SUD literature describing reduced sensitivity to punishment (Simons & Arens, 2007; Simons et al., 2008) as well as affective stimuli more generally (Hester et al., 2013; Stewart et al., 2014). Given that individuals with SUDs also reported greater approach motivation on the task, these findings could index heightened motivation to seek immediate rewards, which may outweigh the expected negative consequences of substance use (such as loss of career or social support). In general, this range of possible interpretations highlights the need for a refined AAC task designed to disambiguate these different factors.

The consistent sex differences we observe, in which it was primarily female HCs that show greater avoidance than those in the clinical groups, may also relate to other known sex differences in affect and decision making. For example, studies have shown that females rely more on avoidant coping mechanisms in general, which in turn positively relates to anxiety (Panayiotou et al., 2017; Sinha & Latha, 2018). However, it should be kept in mind that we found a proportionally higher dropout rate in males over time, which could have influenced results. On the other hand, there were no differences identified between males who did vs. did not return for follow-up (e.g., in affective or substance use symptom severity) that would raise concerns in this regard.

Several novel results in the combined samples also offered insights of potential clinical relevance. For example, we evaluated the utility of our computational model to capture the deliberative aspects of decision-making captured by response times (RTs) and tested for evidence of associated clinical differences. Here we found that correlations between trial-by-trial choice uncertainty and RTs were significantly positive across all participants, but also that this relationship was stronger in SUDs than in the other groups, suggesting potential differences in the deliberation process matching the observed differences in choice. This was further supported by a stronger inverse relationship between RTs and approach behavior in SUDs relative to HCs (i.e., based on both average runway position and *EC*). This relationship to longer RTs appears congruent with the greater *DU* values found in SUDs and suggests a greater contribution of uncertainty in reaching approach decisions (recall that SUDs also showed greater RTs than the other groups in general). Machine learning analyses combining RT-uncertainty correlation values with model parameters (at both time points) could also discriminate between individuals who had only DEP/ANX from those with only SUDs in an out-of-sample manner (balanced accuracy = 0.71). Similarly, this machine learning approach could predict the presence or absence of SUDs (balanced accuracy = 0.69), and the presence or absence of DEP/ANX (balanced accuracy = 0.69) irrespective of comorbidities, as well as classify those with or without comorbidities in general (i.e., between affective and substance use disorders; balanced accuracy = 0.64), potentially reflecting overall illness severity. Thus, overall, the stability of these maladaptive choice patterns over time, and potential differences in deliberative processes, may go beyond the general presence of psychopathology. Instead, it appears they also differentiate between disorders both with and without comorbidity.

Here it should be highlighted that some notable differences in *DU* were found between the exploratory and confirmatory samples. First, both clinical groups (DEP/ANX and SUDs) showed lower *DU* in the confirmatory sample than in the exploratory sample at baseline. Second, HCs at follow-up showed higher *DU* in the confirmatory sample than in the exploratory sample (although

the associated BF did not provide support for this difference). These differences led in part to the further result that HCs and DEP/ANX showed smaller changes over time in the confirmatory sample than in the exploratory sample. This also helped to account for why *DU* did not differ in the expected manner between SUDs and the other two groups at follow-up. On the other hand, both clinical groups showed similar values between samples at follow-up, and previously observed reductions in *DU* over time successfully replicated. In SUDs, the pattern of change over time in *DU* was also very similar between samples and thus supported its replicability. Here, it is also worth noting that sample differences in *DU* were not associated with any other specific sample characteristics, and thus appeared to reflect an independent trait difference in behavior.

Before concluding, it is important to emphasize some remaining limitations. First, as mentioned above, the sample was imbalanced with respect to clinical group membership and sex ratio within each group. There was also differential drop-out over time. That said, our intent-to-treat analysis approach aimed to minimize any biasing effects of drop-out, and machine learning analyses made use of balanced accuracy metrics designed to compensate for imbalanced sample sizes. Next, as our community sample was not intentionally recruited to test the presence vs. absence of psychiatric comorbidities (e.g., generalized anxiety disorder without major depression), future work will be necessary to compare specific affective or substance use diagnoses more definitively. Our recruitment criteria also did not exclude participants based on medication or treatment status, meaning some participants displayed symptom improvement over time. This variable improvement could also stem in part from the fact that the larger T1000 project was observational and naturalistic, and therefore did not allow control over other participant activities between baseline and follow-up. In addition, individuals with substance use disorders were recruited from recovery homes, and were both treatment-seeking and currently abstinent. Each of these issues therefore likely contributed to the average symptom improvement we observed over time between baseline and follow-up. We also cannot rule out possible effects of regression to the mean when considering changes over time. That said, any effects of baseline differences were accounted for in analyses of change scores, which should have minimized this potential confound. It should also be noted that, despite symptom reductions, group-level differences in computational mechanisms were largely conserved – suggesting these may reflect independent trait differences.

Further consideration should also be given to other results that did not replicate, and whether study limitations could relate to this in any way. For instance, in the exploratory sample, we found that baseline *EC* predicted change in BIS scores at 1-year follow-up; but this finding did not replicate in our new sample. Instead, other significant relationships between task measures and dimensional measures were observed in the confirmatory sample that were not present in the exploratory sample. These inconsistencies in findings may indicate the presence of either false positives or false negatives in either the current or previously reported findings; or they could pertain to the baseline group differences in other measures discussed above. In general, these results further support the importance of confirmatory studies. Finally, it should be noted that other modeling approaches could have been used. However, we followed our previous studies (Smith et al., 2021b; Smith et al., 2021c; Smith et al., 2023) and a pre-registered approach, and the model appeared to perform well in capturing decision dynamics.

In summary, these findings contribute to ongoing efforts to replicate and confirm the generalizability of prior results. They also extend the utility of previously observed transdiagnostic markers by demonstrating their further ability to predictively categorize different clinical groups. Not all results replicated, further supporting the importance of replication studies. In particular, the ability of the model parameters to predict symptom changes over time did not appear consistent between studies. In contrast, the results that did replicate, as well as the novel evidence supporting the potential utility of *EC* and *DU* as diagnostic predictors, should now motivate future work aimed at intervening on these mechanisms to evaluate their causal relevance. It should also motivate further work aimed at better understanding underlying neural mechanisms and potential relationships with changes in symptom severity.

## ADDITIONAL FILE

The additional file for this article can be found as follows:

- **Supplementary Materials.** Supplementary Results, Figures, and Tables. DOI: https://doi.org/10.5334/cpsy.131.s1

## FUNDING INFORMATION

This study was supported by the Laureate Institute for Brain Research.

## COMPETING INTERESTS

M.P.P. is an advisor to Spring Care, Inc., a behavioral health startup; has received royalties for an article about methamphetamine in UpToDate; and has a consulting agreement with and receives compensation from F. Hoffmann-La Roche Ltd. No other competing interests were declared.

## AUTHOR CONTRIBUTIONS

M.M.M. and R.S. led the formal analysis and manuscript preparation. N.H contributed to the formal analysis. R.A. led task development, and M.P.P. led the larger funded study in which these data were collected. All authors reviewed, revised, and approved the final manuscript.

## AUTHOR AFFILIATIONS

**Marishka M. Mehta** orcid.org/0000-0001-5105-6723
Laureate Institute for Brain Research, Tulsa, OK, US; University of Tulsa, Tulsa, OK, US

**Navid Hakimi** orcid.org/0000-0002-8624-1838
Laureate Institute for Brain Research, Tulsa, OK, US

**Orestes Pena** orcid.org/0009-0002-5995-8056
Laureate Institute for Brain Research, Tulsa, OK, US

**Taylor Torres** orcid.org/0000-0002-5538-9581
Laureate Institute for Brain Research, Tulsa, OK, US

**Carter M. Goldman** orcid.org/0009-0001-5285-4361
Laureate Institute for Brain Research, Tulsa, OK, US

**Claire A. Lavalley** orcid.org/0009-0005-2994-662X
Laureate Institute for Brain Research, Tulsa, OK, US; University of Tulsa, Tulsa, OK, US

**Jennifer L. Stewart** orcid.org/0000-0003-0123-9438
Laureate Institute for Brain Research, Tulsa, OK, US

**Hannah Berg** orcid.org/0000-0002-8283-0133
Laureate Institute for Brain Research, Tulsa, OK, US

**Maria Ironside** orcid.org/0000-0003-2781-1711
Laureate Institute for Brain Research, Tulsa, OK, US

**Martin P. Paulus** orcid.org/0000-0002-0825-3606
Laureate Institute for Brain Research, Tulsa, OK, US

**Robin Aupperle** orcid.org/0000-0003-2173-6140
Laureate Institute for Brain Research, Tulsa, OK, US

**Ryan Smith** orcid.org/0000-0002-4448-185X
Laureate Institute for Brain Research, Tulsa, OK, US

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
