## [Reviewer Report · Peer Review History]

## Recommendation:

Accept with Minor Revision

## Comments/Explanation (required). Please include references to any minor or major revisions required, as well as addressing any issues with language clarity. Authors will see these comments:

Dear editorial board,

## Peer-review-recommendation

Accept Submission

---

## [Reviewer Report · Peer Review History]

## Recommendation:

Accept with Major Revision

## Comments/Explanation (required). Please include references to any minor or major revisions required, as well as addressing any issues with language clarity. Authors will see these comments:

The aim of the paper seems important to the field and worthy of publication. However, a substantial revision of the manuscript is required. In general, the manuscript's writing and structure are messy, including many different types of exploratory analyses. It is difficult for the reader to keep track. Although all analyses may be warranted, I recommend the authors to restructure the manuscript, including clear explanatory and consistent subheadings throughout both manuscript and supplementary.

Together, this maintains the reader's attention and emphasizes the paper's focus – the key dependent variables of emotional conflict and decision uncertainty.

---

## [Reviewer Report · Peer Review History Round 1, Simon Jangard]

## Reviewer C

1. Given the large set of analyses, I would clearly distinguish the pre-registered and the exploratory analyses in the paper. Additionally, I would consider moving some of the exploratory analyses to the supplementary in order to improve the message of the manuscript.

## Author’s response

In line with this feedback, subheadings have now been added to group the first set of results under “Pre-registered Replication Analyses” and the second set of analyses under “Combined Sample Analyses”. Similar grouping headings have also been provided in the Methods, and figure labels have also been added and updated to include specific language that distinguishes between pre-registered analyses and the additional analyses performed in the combined sample. We appreciate the suggestion to move some exploratory results to the supplementary results. We considered various ways of doing this. However, we ultimately came to the conclusion that some of the strengths and overall narrative of the paper were lost each way we considered. In particular: (1) the importance of pre-registered replication and (2) how this can help identify computational markers with predictive diagnostic utility (itself a central goal that computational psychiatry has been working towards as a field). We also note that a fairly large number of results have already been placed in supplementary materials. Thus, in our view, the manuscript remained stronger by keeping this material in the main text. We believe that the suggested changes we have made to the subheadings and other language now better guides the reader through the overarching narrative. We hope the reviewer agrees, and we are of course willing to consider this further if the reviewer or editor feels strongly about this. Below we quote a few representative changes to the language in the manuscript that exemplify these revisions:

Introduction:

Pg. 5 - “*The current study aimed to address three central questions. First, we tested the replicability of previously reported longitudinal results and observed sex differences in a new community-based sample. These pre-registered analyses (https://osf.io/7hsx9) specifically aimed to replicate: (1a) previously observed reductions in DU over time; as well as (1b) consistently elevated DU over time in both DEP/ANX and SUDs relative to HCs. We further aimed to replicate: (1c) previously observed reductions in EC within each clinical group compared to HCs that were stable over time, where this effect was driven by differences in females but not males. After completing each of these pre-registered analyses, we then combined the prior and current datasets, which afforded greater statistical power to test questions that we were limited in our ability to explore previously. This included examining: (2) whether observed differences by group and sex were driven by narrower diagnostic categories (i.e., for each specific affective and substance use disorder); and (3) whether computational measures afforded predictive utility in distinguishing between diagnoses. To do so, we used state-of-the-art machine learning approaches to test the out-of-sample predictive accuracy of these computational measures in classifying individuals into either the DEP/ANX or SUD groups. This provided a direct test of whether DU and EC levels could jointly contribute to differential diagnosis (i.e., differentiating those with only an affective disorder, the presence of at least one SUD, or both). Together, these analyses aimed to demonstrate whether a computational characterization of behavior under AAC might provide both mechanistic and pragmatic information that could be useful in a clinical setting.*”

Discussion:

Pg. 35 - *“The current study was designed to achieve three main goals. First, as pre-registered, we repeated a set of longitudinal analyses in a new sample of individuals with depression and/or an anxiety disorder (DEP/ANX), substance use disorders (SUDs), and healthy comparisons (HCs) who each completed an approach-avoidance conflict (AAC) task at two time points, one year apart.”*

Pg. 36 - “For our second goal, we combined participant data from the previous and current samples to answer more granular questions about how computational mechanisms might differentiate specific clinical disorders. Namely, we tested whether individuals with each specific affective or substance use disorder would show the same pattern of group differences relative to HCs as seen in the broader groupings. As discussed below, these results supported a largely transdiagnostic pattern. Finally, for our third goal, we carried out a machine learning-based (out-of-sample) classification approach to evaluate whether computational modeling measures could predictively identify individuals as having affective or substance use disorders (or comorbidity of the two), in which results showed moderate levels of accuracy.”

Here, we also provide a full list of figure legends in the paper. We hope this provides further clarity and distinction between results from pre-registered analyses and extended analyses in the combined sample:

*Figure 1. The approach-avoidance (AAC) task*.*Figure 2. Overview of study design and key objectives*.*Figure 3. Comparison of exploratory and confirmatory samples*.*Figure 4. Pre-registered tests of group, time, and sex effects on model parameters*.*Figure 5. Effects of group and sex on emotion conflict (EC) across specific diagnostic categories in the combined sample*.*Figure 6. Effects of group and sex on decision uncertainty (DU) across specific diagnostic categories in the combined sample*.*Figure 7. Predictive classification of diagnostic status in the combined clinical sample*.

Finally, we have included a new figure that summarizes the study protocol and demarcates the key findings from pre-registered aims and extended analyses in the combined sample.

Pg. 19 - *“A schematic summary of the study design, including pre-registered replication aims in the confirmatory 1-year follow-up sample, and subsequent analyses in the combined sample, is presented in Figure 2*.”

**Figure 2 d67e6825:** **Overview of study design and key objectives.** The current study is part of the larger Tulsa 1000 project (Victor et al., 2018). This project collected data from two independent samples, an *exploratory* sample and a *confirmatory* sample (Ns shown in *bottom left* panel), both at a baseline time point and a 1-year follow-up. At each time point, participants completed (among other measures) an Approach Avoidance Conflict (AAC) task and a set of clinical and dimensional scales (*top left* panel). These included the following: the Patient Health Questionnaire (PHQ; depression), the Overall Anxiety Severity and Impairment Scale (OASIS), the Drug Abuse Screening Test (DAST-10), the Anxiety Sensitivity Index (ASI), the Behavioral Activation/Inhibition scales (BIS/BAS), the Patient-Reported Outcomes Measurement Information System (PROMIS) depression and anxiety scales, the Positive and Negative Affect Schedule (PANAS), the State-Trait Anxiety Inventory (STAI), and the Temporal Experience of Pleasure Scale (TEPS). For inclusion in the study, the clinical groups were required to meet specific criteria across the PHQ, OASIS, and DAST-10 (*top left*). Formal diagnostic grouping was then performed in accordance with DSM-IV or DSM-5 criteria using the Mini International Neuropsychiatric Interview (MINI), administered at baseline by trained professionals. Building on prior work with this dataset, the pre-registered aims of the current study focused on 1-year follow-up data in the confirmatory sample (*top right* panel). Namely, we tested the replicability of several longitudinal results previously reported on the exploratory dataset. These centered on to two parameters derived from computationally modeling AAC task behavior: decision uncertainty (*DU*) and emotion conflict (*EC*). Following completion of all pre-registered analyses, we then combined the exploratory and confirmatory samples, affording power to test additional questions (*bottom right* panel). Check marks and Xs indicate which of these analyses were and were not successfully replicated, respectively.

DSM-5 criteria using the Mini International Neuropsychiatric Interview (MINI), administered at baseline by trained professionals. Building on prior work with this dataset, the pre-registered aims of the current study focused on 1-year follow-up data in the confirmatory sample (*top right* panel). Namely, we tested the replicability of several longitudinal results previously reported on the exploratory dataset. These centered on to two parameters derived from computationally modeling AAC task behavior: decision uncertainty (*DU*) and emotion conflict (*EC*). Following completion of all pre-registered analyses, we then combined the exploratory and confirmatory samples, affording power to test additional questions (*bottom right* panel). Check marks and Xs indicate which of these analyses were and were not successfully replicated, respectively.

## Reviewer C

2. Is it a clinical sample or community-based sample? These terms are used interchangeably. From the methods, it seems like it is a community-based sample with clinical symptoms. Please correct so that it is consistent throughout.

## Authors's response

Reply #2: We appreciate how this terminology could be unclear. In fact, we intended to use these terms to refer to different things (not interchangeably). To clarify, this study included data collected through community-based recruitment and included both healthy comparisons (HCs) and individuals who met criteria for one or more disorders. When we used the term ‘community sample’ this was meant to refer to all participants (HCs and participants with one or more disorders). When we used the term ‘clinical sample’, we meant to refer only to participants with one or more disorders. Other than HCs, our community-based screening required only that individuals met criteria for minimum symptom levels of depression, anxiety, or substance use. A subsequent diagnostic interview then evaluated for the presence of one or more disorders. In the present study, we then divided groups based on their diagnoses. Thus, it is a community sample overall, but after recruitment participants were divided into clinical and non-clinical samples. These recruitment procedures were part of the larger T1000 study protocol, which we have been sure to cite. We have also assured that all details mentioned here are now also clearly described in the text. Furthermore, we have modified language to clarify that the term “community-based sample” refers explicitly to the recruitment procedures, while the use of “clinical sample” implies individuals within this community-based sample who met criteria for depression, anxiety, or substance use disorders (i.e., not healthy comparisons) and are grouped as such for the purposes of our analysis plan. For example:

Pg. 7 - “*As in our previous work in an exploratory sample (i.e., taken from the first 500 participants recruited to the T1000 study; Smith et al., 2021b; Smith et al., 2021c), the confirmatory sample (i.e., taken from the subsequent 550 participants recruited to the T1000 study; Smith et al., 2023) was divided into three groups: (i) HCs (baseline: N=97, 1-year follow-up: N=69); (ii) DEP/ANX (baseline: N=208, 1-year follow-up: N=135; including individuals who met criteria for one or more of the following: major depressive disorder, social anxiety disorder, generalized anxiety disorder, panic disorder, and/or posttraumatic stress disorder); (iii) SUDs (baseline: N=175, 1-year follow-up: N=83; with or without comorbid anxiety/depression). A full breakdown of co-morbidities within the clinical sample (i.e., individuals who met criteria for DEP/ANX or SUDs) is provided in Supplementary Table 1*.”

## Reviewer C

3. The subgroup analysis is probably flawed due to the small N in each subgroup. If you decide to keep it, you have to include some power considerations to show that these subgroups are meaningful to analyze in relation to the parameters of interest.

## Authors's response

Reply #3: We appreciate this feedback. We agree that the small N in certain subgroups limited the interpretability and generalizability of some findings. This motivated inclusion of effect sizes for post-hoc comparisons (provided in Supplementary Tables 26-29) and Bayesian t-tests, which allowed us to test for the presence or absence of group differences. We also performed additional sensitivity power analyses to indicate the minimum effect sizes we were powered to detect. The updated methods, results, and associated supplementary table (Supplementary Table 30) reflecting these changes are provided below:

Methods:

Pg. 15 - “*As some subgroups had only moderate sample sizes, we performed post-hoc power analyses by simulation to examine the influence of these varying sample sizes across specific diagnostic categories. These analyses focused on LMEs with the following structure: parameter ~ group*time + (1|participant). Without losing generality, the between- and within-subject variances were set to 0.1 and 0.9, respectively, such that the total variance is 1. Missing values were introduced to the outcome variable with a 0.25 probability for HCs and between 0.32-0.64 probabilities for each clinical group at 1-year follow-up. These probabilities correspond to the attrition rate for the clinical group in question. Group and visit variables were coded as [-1, 1] such that the associated beta coefficient indicated 1/4 of the mean outcome averaged over visits between groups. Using this setup, the data were then used to fit the aforementioned model structure. This process was repeated 4,000 times, where the computed statistical power represents the proportion of p-values < 0.05 for the group term*.”

Results:

Pg. 28 - “*After carrying out all pre-registered analyses above, we then combined exploratory (N_Baseline_=478; N_Follow-up_=324) and confirmatory (N_Baseline_=480; N_Follow-up_=287) samples. This provided greater power to examine whether observed group differences were consistent across the more specific diagnostic categories present within the dataset (e.g., generalized anxiety disorder, social anxiety disorder, methamphetamine use disorder, opioid use disorder, etc.). To answer this question, separate LMEs were performed including only HCs and different subsets of participants from the combined clinical groups. Namely, one LME was performed including the subset of participants who met criteria for each specific disorder (e.g., all those with stimulant [cocaine or methamphetamine] use disorder, or all those with major depressive disorder, etc.). Note that, due to comorbidities, these subsets of participants had partial overlap (e.g. individuals with major depressive disorder may or may not have had other anxiety or substance use disorders). These LMEs then predicted each model parameter based on the specific diagnostic category in question (e.g., HCs versus generalized anxiety, HCs versus opioid use disorder, etc.); effects of time and the interaction term were also included. Subsequent post-hoc power analyses also considered attrition rate within each sample when testing for effects of group (at a 0.05 significance level) on DU and EC, respectively. As a secondary check, these LMEs were also re-performed when including possible effects of age, sex, and their interaction with group. Here we found consistent main effects of group across specific disorders for each parameter (Figures 5-6)*.

…*Group differences were also found in DU, whereby HCs had lower DU than most sub-groups (Fs≥4.08, ps≤0.044). A similar but non-significant pattern was found in panic disorder (F(1, 271)=3.47, p=0.064). Generalized and social anxiety disorders did not show significant effects. Post-hoc simulations suggested power of β>0.846 to detect group differences with the observed effect sizes, with the exception of generalized and social anxiety disorders, which had associated power levels of only 0.584 and 0.086 (i.e., due to the small observed effect sizes). A main effect of time (Fs≥19.80, ps<0.001) was also found, with lower DU at follow-up in all cases*.

…*Descriptive statistics for model parameters in each specific diagnostic group are provided in Supplementary Table 25. Full results of all LMEs are provided in Supplementary Tables 26-29. Results of associated post-hoc power analyses are provided in Supplementary Table 30*.”

**Supplementary Table 30. ST30:** Simulation-based post-hoc power analyses to detect group differences between HCs and clinical subgroups

**Group**	**N at baseline** **(% attrition at follow-up)**	** *DU* **	** *EC* **

Cohen’s *d*	Power	Cohen’s *d*	Power

**MDD**	619 (36.19%)	0.183	0.955	–0.303	1.000
**GAD**	277 (32.13%)	0.115	0.584	–0.271	1.000
**Social**	164 (37.80%)	0.039	0.086	–0.353	1.000
**Panic**	113 (32.74%)	0.199	0.846	–0.310	1.000
**PTSD**	123 (34.96%)	0.243	0.955	–0.245	0.977
**Alcohol**	118 (55.08%)	0.332	0.998	–0.631	1.000
**Cannabis**	158 (56.33%)	0.284	0.987	–0.595	1.000
**Stimulant**	278 (51.44%)	0.322	0.999	–0.599	1.000
**Opioid**	122 (49.18%)	0.317	0.997	–0.572	1.000
**Sedative**	84 (61.90%)	0.281	0.919	–0.671	1.000

## Reviewer C

4. The fact that the symptom levels on all scales (Table 2) are improved from baseline to follow-up is not addressed. What does this mean? This should be addressed in the discussion part of the manuscript.

## Authors's response

Reply #4: We agree it would be helpful to address this more explicitly. Because the T1000 was a naturalistic observational study, there were no restrictions on participants’ activities between baseline and 1-year follow-up. As noted in the methods, participants with SUDs were also recruited from recovery homes and currently abstinent, so we would expect some subsequent improvement on average. Some improvements might also reflect regression to the mean, as those with the lowest scores at baseline would be expected to report less severity upon repeated testing (which is why our analyses of symptom change controlled for baseline levels). We have now mentioned these points more explicitly in the manuscript. For example:

Pg. 40 - *“Before concluding, it is important to emphasize some remaining limitations… Our recruitment criteria also did not exclude participants based on medication or treatment status, meaning some participants displayed symptom improvement over time. This variable improvement could also stem in part from the fact that the larger T1000 project was observational and naturalistic, and therefore did not allow control over other participant activities between baseline and follow-up. In addition, individuals with substance use disorders were recruited from recovery homes, and were both treatment-seeking and currently abstinent. Each of these issues therefore likely contributed to the average symptom improvement we observed over time between baseline and follow-up. We also cannot rule out possible effects of regression to the mean when considering changes over time. That said, any effects of baseline differences were accounted for in analyses of change scores, which should have minimized this potential confound. It should also be noted that, despite symptom reductions, group-level differences in computational mechanisms were largely conserved – suggesting these may reflect independent trait differences*.”

## Reviewer C

5. The clinical scales used in Table 2 have cut-offs. For the unfamiliar (and clinically-oriented) reader, and to highlight the clinical relevance of the study, please provide somewhere what the mean values in each group correspond to (e.g., moderate depression, etc. ), E.g., in Table 2.

## Authors's response

Reply #5: We agree this would help orient readers to the categorical symptom severity of our clinical sample. To address this, we have now updated the legend for Table 2 to include a brief description of the cut-offs and corresponding severity levels for the PHQ, OASIS, and DAST-10 measures. Specifically, we now note that PHQ-9 scores of 0-4 correspond to minimal depression, 5-9 to mild depression, 10-14 to moderate depression, 15-19 to moderately severe depression, and scores above 19 to severe depression; OASIS scores of 8 or above indicate clinically significant anxiety; and DAST-10 scores of 1-2 suggest low levels of substance use-related problems, 3-5 to moderate level, 6-8 to substantial levels, and 9-10 suggest severe levels of impairment.

**Table 2 d67e7157:** Summary statistics and group differences for demographic and clinical measures.

	HCs	DEP/ANX	SUDs	*p**

Baseline	N = 97	N = 208	N = 175	

Follow-up	N = 69	N = 135	N = 83	

**Age**				

Baseline	32.09 (11.10)	33.74 (10.17)	33.76 (8.42)	0.33

Follow-up	34.50 (11.30)	34.69 (10.17)	34.41 (8.56)	0.978

**Sex (Male)** ^+^				

Baseline	38 (39.20)	51 (24.50)	68 (38.90)	0.004**

Follow-up	23 (33.30)	29 (21.50)	22.00 (26.50)	0.184

**PHQ-9**				

Baseline	1.23 (1.82)	12.53 (5.00)	6.59 (5.76)	<0.001***

Follow-up	0.91 (1.73)	8.47 (5.62)	3.30 (4.44)	<0.001***

**OASIS**				

Baseline	1.10 (1.62)	9.65 (3.58)	5.77 (4.46)	<0.001***

Follow-up	0.86 (1.75)	7.11 (4.36)	3.26 (3.73)	<0.001***

**DAST-10**				

Baseline	0.19 (0.49)	0.41 (0.95)	7.46 (2.23)	<0.001***

Follow-up	0.30 (0.55)	0.37 (0.68)	2.27 (2.53)	<0.001***

**WRAT reading score**				

Baseline	62.81 (5.31)	62.75 (5.06)	56.79 (6.78)	<0.001***

Follow-up	63.25 (5.63)	63.02 (5.03)	57.12 (7.12)	<0.001***

**Note:** For ease of interpretation, commonly used severity cutoffs for each symptom measure are as follows. Patient Health Questionnaire (PHQ): 0–4 indicates minimal depression symptoms, 5–9 indicates mild depression, 10–14 indicates moderate depression, 15–19 indicates moderately severe depression, and scores > 19 indicate severe depression (30). Overall Anxiety Severity and Impairment scale (OASIS): ≥ 8 is suggestive of clinically significant anxiety (more specific severity ranges are not provided; [8]). Drug Abuse Screening Test (DAST): 1–2 indicates low levels of impairment, 3–5 indicates moderate levels of severity, 6–8 indicates substantial levels of severity, and scores of 9–10 indicates severe levels of severity (54). Please note that not all individuals in the DEP/ANX group displayed elevated symptoms for both anxiety and depression (i.e., had co-morbid anxiety and depression) as measured by OASIS and PHQ, respectively.WRAT = Wide Range Achievement Test; **p*-values are based on ANOVAs testing for significant differences between the three groups (****p* < 0.001; ***p* < 0.01; **p* < 0.05); ^+^Sex is reported in terms of counts and percentages. Reported *p*-values are based on Chi-Squared tests.

## Reviewer C

6. To address research questions 2 and 3 in the final paragraph of the introduction (i.e., testing narrower diagnostic groups and out-of-sample predictive accuracy), I would suggest that you include some power considerations. I would relate it to previous studies in the field in terms of sample size (including the subgroups) and expected effect sizes using the same or similar types of parameter.

## Authors's response

Reply #6: We agree it would be helpful to more explicitly address power for these analyses, given the varying sample sizes by group and analysis. With respect to research question 2, and as also described in Reply #3, we have now run power analyses for each associated analysis and shown sufficient power to detect meaningful effect sizes, now detailed in Supplementary Table 30 (reproduced below for convenience).

With respect to research question 3, we are unaware of a standard approach for calculating power for the out-of-sample prediction approach we employed. However, the use of ensemble models with up-sampling, 5-fold cross-validation, and balanced accuracy aligns well with best practices for handling class imbalance and over-fitting issues that may arise from class imbalance (Khan et al., 2024; Thölke et al., 2023). Similar studies using neurobehavioral data have demonstrated successful classification with comparable class ratios and sample sizes. For example, Lee et al. (2025) used ERP data to estimate suicide risk by predictively distinguishing attempters (N=22) from non-attempters (N=35) with AUCs ranging from 0.78 to 0.89. Similarly, another study (N=102) reported that a classification model could correctly predict PTSD and SUD co-morbidity in 86.67% of participants (Houghton et al., 2023). In general, few studies have focused on binary classification; however, for a detailed review on use of neurobehavioral data for outcome predictions, see Aafjes-van Doorn et al. (2021)). Our achieved AUC of 0.77 and balanced accuracy of 0.69 (and by extension, the comparable AUC and balanced accuracy values within other classifications within the paper) suggest good model performance. Therefore, we are confident in the reliability of our results. Nonetheless, we do not make any strong statements interpreting null results. In addition to the Table, we also reproduce revised language below that ensures each of these points are clear and incorporates this prior work.

Pg. 18 - “*The 5-fold cross-validation consisted of first dividing the dataset into five equal subsets. For each of the five rounds, four subsets were combined and used as a training dataset and the fifth was used as a testing dataset. In each round, the training dataset was also further divided into training (80%) and validation (20%) subsets. The validation subset was used for finding optimal hyperparameters. Here, seven different starting values were randomly selected for each hyper-parameter and a grid-search approach was used to find the optimal values. After parameter tuning was complete, the stacked ensemble approach then used linear aggregation to combine the predictions of each algorithm. Specifically, the weight on each algorithm was determined by respective model fit (area under the curve [AUC]), such that algorithms with better fits to the data were afforded higher weight in generating the stacked prediction. This process was repeated five times to get a stable comparison of fitted stacked models (STACK), and we report the average performance across these 5 folds. Following best practices for evaluating models on imbalanced datasets (Thölke et al., 2023), we used AUC and balanced accuracy (arithmetic mean of sensitivity and specificity) as performance metrics. Previous work using similar approaches in clinical samples is limited, but available studies have found AUCs ranging between 0.7-0.9 and balanced accuracies around 0.7 (Aafjes-van Doorn et al., 2021; Houghton et al., 2023; Lee et al., 2025); thus, we hypothesized similar levels of performance. Sensitivity and specificity, which refer to the true positive and negative rate, respectively, are also reported in results. In addition, a variable importance (VI) metric was computed to measure the average scaled contribution (between 0 to 100) of each predictor. As noted above, due to the unbalanced sample size, an up-sampling approach was also used during the cross-validation process*.”

**Supplementary Table 30. d67e7542:** Simulation-based post-hoc power analyses to detect group differences between HCs and clinical subgroups

**Group**	**N at baseline** **(% attrition at follow-up)**	** *DU* **	** *EC* **

Cohen’s *d*	Power	Cohen’s *d*	Power

**MDD**	619 (36.19%)	0.183	0.955	–0.303	1.000
**GAD**	277 (32.13%)	0.115	0.584	–0.271	1.000
**Social**	164 (37.80%)	0.039	0.086	–0.353	1.000
**Panic**	113 (32.74%)	0.199	0.846	–0.310	1.000
**PTSD**	123 (34.96%)	0.243	0.955	–0.245	0.977
**Alcohol**	118 (55.08%)	0.332	0.998	–0.631	1.000
**Cannabis**	158 (56.33%)	0.284	0.987	–0.595	1.000
**Stimulant**	278 (51.44%)	0.322	0.999	–0.599	1.000
**Opioid**	122 (49.18%)	0.317	0.997	–0.572	1.000
**Sedative**	84 (61.90%)	0.281	0.919	–0.671	1.000

## Reviewer C

7. Related to above. I suggest that you be more consistent in the use of subheadings to guide the reader. E.g., if you have four main research questions in your manuscript that you are addressing: 1) replicating earlier longitudinal results (BTW, this could be specified in more detail in the last paragraph of the intro); 2) replicating earlier sex differences; 3) Explore narrower diagnostic groups; and 4) testing out-of-sample predictive accuracy. Repeat relevant keywords in the main subheadings throughout the methods and results. At the current moment, I have to go back-and-forth many times to understand which research questions you are addressing.

## Authors's response

Reply #7: We appreciate this suggestion and agree more consistency in subheadings could increase readability. As also detailed in Reply #1, we have now revised the manuscript to ensure that each major research question is clearly stated and consistently highlighted throughout the methods and results sections. We also appreciate the reviewer’s suggested division of our main research questions and have adopted them. However, we consider the first two questions to overlap enough within our analyses and results that we combine them and frame our results in terms of three main research questions. Some of the associated changes are re-stated below for convenience.

Introduction:

Pg. 5 - “*The current study aimed to address three central questions. First, we tested the replicability of previously reported longitudinal results and observed sex differences in a new community-based sample. These pre-registered analyses (https://osf.io/7hsx9) specifically aimed to replicate: (1a) previously observed reductions in DU over time; as well as (1b) consistently elevated DU over time in both DEP/ANX and SUDs relative to HCs. We further aimed to replicate: (1c) previously observed reductions in EC within each clinical group compared to HCs that were stable over time, where this effect was driven by differences in females but not males. After completing each of these pre-registered analyses, we then combined the prior and current datasets, which afforded greater statistical power to test questions that we were limited in our ability to explore previously. This included asking: (2) whether observed differences by group and sex were driven by narrower diagnostic categories (i.e., for each specific affective and substance use disorder); and (3) whether computational measures afforded predictive utility in distinguishing between diagnoses. To do so, we used state-of-the-art machine learning approaches to test the out-of-sample predictive accuracy of these computational measures in classifying individuals into either the DEP/ANX or SUD groups. This provided a direct test of whether DU and EC levels could jointly contribute to differential diagnosis (i.e., differentiating those with only an affective disorder, the presence of at least one SUD, or both). Together, these analyses aimed to demonstrate whether a computational characterization of behavior under AAC might provide both mechanistic and pragmatic information that could be useful in a clinical setting*.”

Discussion:

Pg. 35 - *“The current study was designed to achieve three main goals. First, as pre-registered, we repeated a set of longitudinal analyses in a new sample of individuals with depression and/or an anxiety disorder (DEP/ANX), substance use disorders (SUDs), and healthy comparisons (HCs) who each completed an approach-avoidance conflict (AAC) task at two time points, one year apart*.”

Pg. 36 - *“For our second goal, we combined participant data from the previous and current samples to answer more granular questions about how computational mechanisms might differentiate specific clinical disorders. Namely, we tested whether individuals with each specific affective or substance use disorder would show the same pattern of group differences relative to HCs as seen in the broader groupings. As discussed below, these results supported a largely transdiagnostic pattern. Finally, for our third goal, we carried out a machine learning-based (out-of-sample) classification approach to evaluate whether computational modeling measures could predictively identify individuals as having affective or substance use disorders (or comorbidity of the two), in which results showed moderate levels of accuracy*.”

Please see our response to Reply #1 for further details on changes in the figure legends. Note that we have also updated the *Combined Sample Analyses* subheading within the Methods section to include *Transdiagnostic generalizability* and *Predictive utility* as explicit subheadings within the section on further combined sample analyses.

## Reviewer C

8. Methods. You conduct a “transdiagnostic population screening using e.g. PHQ-9 > 10 as cut-off. However, in Table 2, you report that the e.g. Dep/Anx group have PHQ = 12.53 at baseline with an SD of 5. This does not make sense to me (some people are below 10, although this was the cutoff? Please clarify in the text. The same comment goes for the other pre-screening scales.

## Authors's response

Reply #8: This is a very understandable source of possible misunderstanding, and we appreciate the reviewer bringing it to our attention. This is due to the multi-step screening procedures in the T1000 study. To better explain, all participants completed pre-screening surveys, including PHQ-9, OASIS, and DAST-10. Based on these results, individuals who met the general inclusion and exclusion criteria for the study were invited for an in-depth screening visit, which included the MINI diagnostic interview. This meant that, for example, a person could meet the cutoff for OASIS only (e.g., have a PHQ < 10), and yet subsequently receive a diagnosis of depression. Because we group participants based on diagnoses, some individuals in the depressed group could have lower current symptom severity (and therefore lower PHQ scores). We have now made further efforts to prevent this misunderstanding, such as in the following revised text:

Pg. 6 - “*This study leveraged a community-based sample (N=1050) from the Tulsa 1000 (T1000) study (Victor et al., 2018), which was recruited through radio, electronic media, treatment center referrals, and word of mouth. All participants were between the ages of 18–55 years. This study recruited both healthy comparisons (HCs) and a transdiagnostic clinical group. Within the latter group, individuals with SUDs were recruited from community recovery homes and were currently abstinent. Recruitment for the clinical group began with a transdiagnostic population screening, where participants were required to meet symptom severity cutoffs on at least one of the following dimensional measures: the Patient Health Questionnaire (i.e., depression severity [PHQ];Kroenke et al., 2001) ≥ 10, the Overall Anxiety Severity and Impairment Scale (OASIS; Norman et al., 2006) ≥ 8, and the Drug Abuse Screening Test (DAST-10; Bohn et al., 1991) ≥ 3. For individuals who met one or more of these initial cutoffs, an in-depth diagnostic interview was performed at baseline in accordance with DSM-IV or DSM-5 criteria using the Mini International Neuropsychiatric Inventory (MINI; Sheehan et al., 1998), administered at the baseline visit by a trained professional. Participant diagnostic groupings into those with depression, anxiety, and/or substance use disorders (based on the MINI) at baseline were retained at 1-year follow-up, irrespective of change in self-report symptoms (while analyses assessing change in symptoms over time tested for relevant potential effects; see Supplementary Figure 1). HCs did not show elevated symptoms or meet criteria for any psychiatric diagnosis. Exclusion criteria included a positive test for drugs of abuse; diagnosis of psychotic, bipolar, or obsessive-compulsive disorders; or reported history of moderate to severe traumatic brain injury, neurologic disorders, or severe or unstable medical conditions, active suicidal intent, or plan, or change in medication dose within 6 weeks. For detailed inclusion and exclusion criteria, see Victor et al. (2018)*.

*As in our previous work on the exploratory sample (i.e., taken from the first 500 participants recruited to the T1000 study; Smith et al., 2021b; Smith et al., 2021c), the confirmatory sample (i.e., taken from the subsequent 550 participants recruited to the T1000 study; Smith et al., 2023) was divided into three groups: (i) HCs (baseline: N=97, 1-year follow-up: N=69); (ii) DEP/ANX (baseline: N=208, 1-year follow-up: N=135; including individuals who met criteria for one or more of the following: major depressive disorder, social anxiety disorder, generalized anxiety disorder, panic disorder, and/or posttraumatic stress disorder); (iii) SUDs (baseline: N=175, 1-year follow-up: N=83; with or without comorbid anxiety/depression). A full breakdown of co-morbidities within the clinical sample (i.e., individuals who met criteria for DEP/ANX or SUDs) is provided in Supplementary Table 1. Here it is also worth highlighting that symptom cutoffs and diagnostic grouping did not always perfectly align, as individuals who met criteria for a given disorder (e.g., depression) may not have indicated elevated symptoms for that disorder on self-report measures (e.g., PHQ < 10), but nonetheless were screened into the study because they met the cutoff for another measure (e.g., OASIS ≥ 8).”*

## Reviewer C

9. You mention that diagnostic grouping of HC, DEP/ANX, and SUD were performed by a trained professional at the baseline visit. Good! However, this should be stated much earlier in the methods for clarification, i.e., before this statement: “Healthy individuals did not show elevated symptoms or meet criteria for any psychiatric diagnosis.”

## Authors's response

Reply #9: In our revised methods, we have now adjusted the order of these statements in line with the reviewer’s suggestion:

Pg. 6 - *“This study leveraged a community-based sample (N=1050) from the Tulsa 1000 (T1000) study (Victor et al., 2018), which was recruited through radio, electronic media, treatment center referrals, and word of mouth. All participants were between the ages of 18–55 years. This study recruited both healthy comparisons (HCs) and a transdiagnostic clinical group. Within the latter group, individuals with SUDs were recruited from community recovery homes and were currently abstinent. Recruitment for the clinical group began with a transdiagnostic population screening, where participants were required to meet symptom severity cutoffs on at least one of the following dimensional measures: the Patient Health Questionnaire (i.e., depression severity [PHQ]; Kroenke et al., 2001) ≥ 10, the Overall Anxiety Severity and Impairment Scale (OASIS; Norman et al., 2006) ≥ 8, and the Drug Abuse Screening Test (DAST-10; Bohn et al., 1991) score ≥ 3. For individuals who met one or more of these initial cutoffs, an in-depth diagnostic interview was performed in accordance with DSM-IV or DSM-5 criteria using the Mini International Neuropsychiatric Inventory (MINI; Sheehan et al., 1998), administered at the baseline visit by a trained professional. Participant diagnostic groupings into those with depression, anxiety, and/or substance use disorders (based on the MINI) at baseline were retained at 1-year follow-up, irrespective of change in self-reported symptoms (while analyses assessing change in symptoms over time tested for relevant potential effects; see Supplementary Figure 1). HCs did not show elevated symptoms or meet criteria for any psychiatric diagnosis. Exclusion criteria included a positive test for drugs of abuse; diagnosis of psychotic, bipolar, or obsessive-compulsive disorders; or reported history of moderate to severe traumatic brain injury, neurologic disorders, or severe or unstable medical conditions, active suicidal intent, or plan, or change in medication dose within 6 weeks. For detailed inclusion and exclusion criteria, see Victor et al. (2018)*.”

## Reviewer C

10. Methods. You report that you use e.g., the following measures at baseline and follow-up: “(PROMIS) depression and anxiety scales), etc.. However, in table 2 you report PHQ-9 at baseline and follow-up. Please clarify why and how you have chosen to use different scales. This is not clear.

## Authors's response

Reply #10: We apologize for the confusion. The central symptom measures – PHQ-9, OASIS, and DAST-10 –formed the basis of the pre-screening and were the primary focus of several analyses. However, the T1000 project also collected additional survey measures for more in-depth phenotyping. These were treated as secondary measures in our prior work and used in a largely exploratory manner to assess possible relationships with model parameters and their change over time (Smith et al., 2021c). Thus, in the current paper, we aimed to test the replicability of previously reported findings with these measures. We have adjusted the text to better differentiate primary and secondary measures. We have also been sure to more clearly point readers to locations in Supplementary Materials where equivalent descriptive information is provided for these secondary measures. For example:

Pg. 8 - *“The T1000 protocol included intensive assessment of demographic, clinical, and psychiatric features, with a focus on negative and positive affect, arousal, and cognitive functioning. The complete list of assessments, along with their validity and reliability, are reported elsewhere (Victor et al., 2018). Here, we focus mainly on the symptom measures gathered at screening (PHQ, OASIS, and DAST-10). However, our previous study did also explore potential associations with some of the other available assessments (Smith et al., 2021c). Thus, we also aimed to replicate those results here. These secondary measures included: the Patient-Reported Outcomes Measurement Information System (PROMIS) depression and anxiety scales (Cella et al., 2010), the Behavioral Activation/Inhibition scales (BIS/BAS; Carver & White, 1994), the Positive and Negative Affect Schedule (PANAS; Watson et al., 1988), the Anxiety Sensitivity Index (ASI; Sandin et al., 2001), the Temporal Experience of Pleasure Scale (TEPS; Gard et al., 2006), and the State-Trait Anxiety Inventory (STAI; Spielberger et al., 1970). Use of these dimensional measures aligned with the broader goals of the T1000 project, which was designed around the NIMH Research Domain Criteria (RDoC). All data collection procedures were approved by the Western Institutional Review Board. All participants provided written informed consent before completion of the study protocol, in accordance with the Declaration of Helsinki, and were compensated for participation (ClinicalTrials.gov identifier: #NCT02450240).”*

## Reviewer C

11. Methods. The ACC task has been used in several previous lines of work (the first one published in 2011). For transparency and replicability, please clarify (e.g., in supplementary) whether and how you have updated the task for the current manuscript.

## Authors's response

Reply #11: We agree this would be helpful. Various small changes to the task were made in later papers since the original version presented by Aupperle et al. (2011), such as the number of trials included, the number of starting positions used for the avatar, and the presence of a condition in which reward was present in the absence of threat (added in Aupperle et al, 2015). We have now added a clarifying note to the task description to clarify the relevant differences in the version we used:

Pg. 8 - *“The AAC task (Figure 1) used here has been described in detail in our previous work (Aupperle et al., 2015; Aupperle et al., 2011; Smith et al., 2021b).*

*… Here it should be noted that the AAC task used in the T1000 studies differs in a few respects from the original version described by Aupperle et al. (2011). First, the original task did not include the APP condition, which was introduced in a later adaptation by Aupperle et al. (2015), and has been used in all subsequent versions of the task. Second, the T1000 version included a reduced number of trials (60 total; 12 trials per condition), compared to the 90-trial format used in earlier versions of the task. Due to this shorter design, the avatar's starting position was also restricted to either the center or the end of the runway (left or right), counterbalanced across conditions. In contrast, the 90-trial version allowed the avatar to begin from any position on the runway, permitting two trials per starting position within each condition.”*

## Reviewer C

12. Results. Figure 2, top left panel. There is a significant difference between exploratory and confirmatory samples in decision uncertainty. This should be clearly discussed and influence the conclusions drawn from the findings.

## Authors's response

Reply #12: We thank the reviewer for highlighting the need to address this more specifically. In response, we have now included more explicit discussion of this result. We also discuss how this difference may have influenced replication, or lack thereof, of our previously reported findings. We believe this addition strengthens the interpretation and contextualization of our results. Here is a primary example of relevant added language:

Pg. 39 - *“…some notable differences in DU were found between the exploratory and confirmatory samples. First, both clinical groups (DEP/ANX and SUDs) showed lower DU in the confirmatory sample than in the exploratory sample at baseline. Second, HCs at follow-up showed higher DU in the confirmatory sample than in the exploratory sample (although the associated BF did not provide support for this difference). These differences led in part to the further result that HCs and DEP/ANX showed smaller changes over time in the confirmatory sample than in the exploratory sample. This also helped to account for why DU did not differ in the expected manner between SUDs and the other two groups at follow-up. On the other hand, both clinical groups showed similar values between samples at follow-up, and previously observed reductions in DU over time successfully replicated. In SUDs, the pattern of change over time in DU was also very similar between samples and thus supported its replicability. Here, it is also worth noting that sample differences in DU were not associated with any other specific sample characteristics, and thus appeared to reflect an independent trait difference in behavior.”*

## Reviewer C

13. Results. The intra-class correlations for EC and DU parameters are fair-to-good between baseline and follow-up. What is expected in the field from previous studies? What did you expect prior to conducting the study? Should be stated and discussed in intro and/or discussion.

## Authors's response

Reply #13: Our previous paper (Smith et al., 2021c) examined these ICCs within the exploratory sample, showing fair levels of consistency for both model parameters (0.46 and 0.52 for *DU* and *EC*, respectively). In the present study of the confirmatory sample, ICCs were fair-to-good (0.57 and 0.70, respectively). To our knowledge, there are no other computational modeling studies testing stability of parameters over long timescales comparable to 1 year. There are only test-retest reliability studies over days to weeks, which show widely variable ICCs ranging from poor to excellent depending on the task. To better address this, the updated discussion section confirms that the observed ICCs were in the fair-to-good range, consistent with our prior findings and suggesting that *EC* and *DU* capture relatively stable constructs. We have also briefly highlighted the variable level of test-retest reliabilities over short time periods found in prior work. For example:

Pg. 35 - *“The current study was designed to achieve three main goals. First, as pre-registered, we repeated a set of longitudinal analyses in a new sample of individuals with depression and/or an anxiety disorder (DEP/ANX), substance use disorders (SUDs), and healthy comparisons (HCs) who each completed an approach-avoidance conflict (AAC) task at two time points, one year apart. This allowed us to evaluate the reliability of computational task measures and the stability of group differences over time that were previously observed. These analyses successfully replicated some, but not all, hypothesized results*.

*With respect to reliability, ICCs showed fair-to-good levels of consistency in both model parameters over a 1-year period, consistent with the fair levels of consistency found in our exploratory study (Smith et al., 2021c). To our knowledge, no prior computational modeling studies have tested parameter reliability in other tasks over a comparable time period; but these results appear to fall near the middle of a wide range of test-retest reliability levels (from poor to excellent) found for other computational task measures over significantly shorter time periods (Brown et al., 2020; Chung et al., 2017; Enkavi et al., 2019; Moutoussis et al., 2019; Price et al., 2019; Shahar et al., 2019). These results also support our previous examination of test-retest reliability for EC and DU over a shorter 2-3 week period (Smith et al., 2021c), which demonstrated superior test-retest reliability for the former (ICC=0.84) and similar reliability of the latter (ICC=0.54) relative to the longer time period we tested here.”*

## Reviewer C

14. Results. Results reported under “diagnostic effects over time” need simplifications, summary statements, and/or additional tables to improve readability. It is difficult to digest at the moment.

## Authors's response

Reply #14: To improve overall readability in line with this comment, we have revised the title of this section to *Replication of longitudinal stability* and interactions with sex, which is now organized under the *Pre-registered Replication Analyses* section of the results. We have also included several summary statements to highlight the key takeaways and how these findings relate to one another. We hope this makes it clearer to future readers.

Pg. 24 - *“Replication of longitudinal stability and interactions with sex*

*To evaluate stability in model parameters, LMEs predicting each parameter were estimated including main effects of age, sex, group, and time, as well as interactions between group and sex and group and age (see Table 6). Most notably, the LME predicting DU showed significant main effects of time (F(1, 346)=32.83, p<0.001), and group (F(2, 469)=3.08, p=0.047). This indicated a reduction in DU over time (EMM_Baseline_=1.03; EMM_Follow-up_=0.73), and greater DU in SUDs (EMM=1.05) than the DEP/ANX group (EMM=0.84)*.

*In the LME predicting EC, there was a main effect of group (F(2, 470)=15.86, p<0.001), reflecting greater values in HCs (EMM=1.07) than the clinical groups (EMM_DEP/ANX_ =0.85; EMM_SUDs_=0.44). Further, SUDs had significantly lower EC than DEP/ANX. There was a main effect of sex (F(1, 471)=6.24, p=0.013), whereby females displayed greater EC than males (EMM_Female_=0.80; EMM_Male_=0.66). A group x sex interaction (F(2, 472)=5.41, p=0.048) was also present. This interaction indicated that the group effect was primarily driven by females (Females: EMM_HCs_=1.21; EMM_DEP/ANX_=0.90; EMM_SUDs_=0.41; and Males: EMM_HCs_=0.72; EMM_DEP/ANX_=0.72; EMM_SUDs_=0.53; see Figure 4).*

*Together, these results replicated the following from previous work: (1) reduction in DU over time; (2) greater DU in SUDs than DEP/ANX (although differences with HCs did not replicate); (3) lower EC in both clinical groups than HCs (lowest in SUDs); (4) lower EC in males than females; (5) group differences in EC were driven by females. A further replication summary is provided in Supplementary Figure 4*.

*Similar results were also found in pre-registered LMEs including only effects of group and time, and in further LMEs controlling for possible effects of premorbid cognitive ability (WRAT reading scores) in those with available data (Supplementary Tables 8-9). Analogous LMEs were also performed when only including data from participants who returned for the follow-up visit, again with similar results (Supplementary Figure 5)*.

*Identical analyses were also performed for the descriptive task measures: RTs, average chosen runway position, and choice variability (Supplementary Tables 10-21; Supplementary Figure 6). These results also replicated many of our previously reported findings (detailed in Supplementary Figure 7). Briefly, we observed lower choice variability and faster RTs over time across all groups (main effect of time), as well as reduced avoidance behavior in SUDs (reflected in the average chosen runway position for AV, CONF2, CONF4, and CONF6 trials), which replicated prior results. Consistent with results for EC above, a group x sex interaction was present in the CONF2 and CONF4 conditions, wherein reduced avoidance behavior (as measured by average chosen runway position) in SUDs was observed in female participants only*.

*In contrast, when testing the ability of baseline model parameters or other behavioral task measures to predict change in symptoms over time, most previously reported findings did not replicate (Supplementary Figure 1). The only replicable finding was a negative relationship between average chosen runway position in the CONF2 condition at baseline and change in self-reported BIS scores at follow-up (r=-0.14; p=0.047). Other significant associations were observed (i.e., present in the confirmatory sample but not in the exploratory sample), however, suggesting the presence of either false positives or false negatives in either sample*.

*Lastly, LMEs were also performed for post-task self-report survey items assessing anxiety (Q2), decision-making difficulty (Q3), approach motivation (Q4) and avoidance motivation (Q5; Supplementary Tables 22-24). Results replicated previous findings in some cases, but not others (for details, see Supplementary Figure 8). Successful replications included main effects of group on approach (F(2, 468)=4.87, p = 0.008) and avoidance (F(2, 467)=4.18, p=0.016) motivation, where SUDs (EMM_Approach_=5.44; EMM_Avoid_=2.39) displayed greater approach motivation (EMM_SUDs_=5.44; EMM_HCs_=4.32; t(452.58)=-4.03, p<0.001, d=-0.73) and reduced avoidance motivation (EMM_SUDs_=2.34; EMM_HCs_=3.31; t(448.86)=3.81, p<0.001, d=0.63) compared to HCs. The previously observed group x sex interaction (F(2, 469)=4.32, p=0.014) in avoidance motivation was also observed here, suggesting that the lower avoidance motivation observed in SUDs (EMM_Female_=2.20; EMM_Male_=2.67) than HCs (EMM_Female_=3.64; EMM_Male_=2.56) was driven by females (Females: t(438.38)=4.49, p<0.001, d=0.93; Males: t(486.53)=-0.254, p=0.799, d=-0.07)*.

*In summary, similar to our results above, reduced avoidance motivation within SUDs, specifically within female participants, was consistent when measured using the post-task self-report questionnaire. Together, these results therefore robustly support the previously observed group and sex differences in EC. They also motivate further investigation into the potential clinical relevance of these findings. Having performed each of these pre-registered replication analyses, we next combined the exploratory and confirmatory samples to take advantage of the added power this provided to answer additional questions, which focused on testing the transdiagnostic generalizability and predictive utility of our computational measures*.”

**Table 6 d67e7983:** Results of linear mixed effects models predicting DU and EC in data including all baseline participants, when accounting for effects of group, time, age, and sex.

PREDICTOR*	TEST, *P* VALUE	EMM	POST-HOC CONTRASTS

**Decision Uncertainty (*DU*)**

Group	*F*(2, 469) = 3.08, *p* = 0.047	D/A = 0.84; HCs = 0.88; SUDs = 1.05	D/A – HCs: *t*(441.22) = –0.387, *p* = 0.699, *d* = –0.07D/A – SUDs: *t*(472.34) = -2.23, *p* = 0.026, *d* = –0.33HCs – SUDs: *t*(453.13) = –1.441, *p* = 0.150, *d* = –0.26

Time	*F*(1, 346) = 32.83, *p* < 0.001	T1 = 1.03; T2 = 0.73	T1 – T2: *t*(346.52) = 5.73, *p* < 0.001, *d* = 0.46

Age	*F*(1, 462) = 28.76, *p* < 0.001		

Sex	*F*(1, 470) = 3.23, *p* = 0.073		NS

Group × Age	*F*(2, 467) = 2.84, *p* = 0.060		NS

Group × Sex	*F*(2, 471) = 0.14, *p* = 0.873		NS

**Emotion Conflict (*EC*)**

Group	*F*(2, 470) = 15.86, *p* < 0.001	D/A = 0.85; HCs = 1.07; SUDs = 0.44	D/A – HCs: *t*(448.69) = -2.468, *p* = 0.014, d = –0.49D/A – SUDs: *t*(473.21) = 5.44, *p* < 0.001, d = 0.92HCs – SUDs: *t*(458) = 6.793, *p* < 0.001, d = 1.41

Time	*F*(1, 333) = 0.16, *p* = 0.688		NS

Age	*F*(1, 465) = 5.28, *p* = 0.022		

Sex	*F*(1, 471) = 6.24, *p* = 0.013	Female = 0.80; Male = 0.66	Female – Male: *t*(475.87) = 2.03, *p* = 0.043, *d* = 0.34

Group × Age	*F*(2, 473) = 1.64, *p* = 0.194		NS

Group × Sex	*F*(2, 472) = 5.41, *p* = 0.005	Female:D/A = 0.90; HCs = 1.21; SUDs = 0.41Male:D/A = 0.72; HCs = 0.72; SUDs = 0.53	Female:D/A – HCs: *t*(443.07) = –2.891, *p* = 0.004, *d* = –0.71D/A – SUDs: *t*(464.91) = 5.515, *p* < 0.001, *d* = 1.13HCs – SUDs: *t*(450.94) = 7.009, *p* < 0.001, *d* = 1.83Male:D/A – HCs: *t*(467.08) = –0.018, *p* = 0.986, *d* = –0.01D/A – SUDs: *t*(492.88) = 1.403, *p* = 0.161, *d* = 0.43HCs – SUDs: *t*(483.56) = 1.311, *p* = 0.191, *d* = 0.44

**Note:** HCs = Healthy Comparisons; D/A = Depression and Anxiety; SUDs = Substance Use Disorders; T1 = Baseline; T2 = Follow-up; NS = non-significant.For interpretability, age and WRAT scores were centered; sum coding was used for sex (female = –1; male = 1), time (baseline = –1; follow-up = 1), and group (with HCs coded as –1).

## Reviewer C

15. Results. When testing the ability of baseline model parameters to predict in change in symptoms over time, most previously reported findings did not replicate”. This seems like a striking finding. This should be highlighted more clearly in the discussion.

## Authors's response

Reply #15: We agree it is important to better highlight the relevance of this result. The updated limitations paragraph in the discussion section now more explicitly addresses this. We especially highlight how these findings underscore the importance of replication studies.

Pg. 40 - “*Before concluding, it is important to emphasize some remaining limitations. First, as mentioned above, the sample was imbalanced with respect to clinical group membership and sex ratio within each group. There was also differential drop-out over time. That said, our intent-to-treat analysis approach aimed to minimize the biasing effects of drop-out, and machine learning analyses made use of balanced accuracy metrics designed to compensate for imbalanced sample sizes. Next, as our community sample was not intentionally recruited to test the presence versus absence of some psychiatric comorbidities (e.g., generalized anxiety disorder without major depression), future work will be necessary to compare specific affective or substance use diagnoses more definitively*.

*Our recruitment criteria also did not exclude participants based on medication or treatment status, meaning some participants displayed symptom improvement. This variable improvement could also stem in part from the fact that the larger T1000 project was observational and naturalistic, and therefore did not allow control over other participant activities between baseline and follow-up. In addition, individuals with substance use disorders were recruited from recovery homes and were both treatment-seeking and currently abstinent. Each of these issues therefore likely contributed to the average symptom improvement we observed over time between baseline and follow-up. We also cannot rule out possible effects of regression to the mean when considering changes over time. That said, any effects of baseline differences were accounted for in analyses of change scores, which should have minimized this potential confound. It should also be noted that, despite symptom reductions, group-level differences in computational mechanisms were largely conserved – suggesting these may reflect independent trait differences*.

*Further consideration should also be given to other results that did not replicate, and whether study limitations could relate to this in any way. For instance, in the exploratory sample, we found that baseline EC predicted change in BIS scores at 1-year follow-up; but this finding did not replicate in our new sample. Instead, other significant relationships between task measures and dimensional measures were observed in the confirmatory sample that were not present in the exploratory sample. These inconsistencies in findings may indicate the presence of either false positives or false negatives in either the current or previously reported findings; or they could pertain to the baseline group differences in other measures discussed above. In general, these results further support the importance of confirmatory studies. Finally, it should be noted that other modeling approaches could have been used. However, we followed our previous (Smith et al., 2021b; Smith et al., 2021c; Smith et al., 2023) and a pre-registered approach, and the model appeared to perform well in capturing decision dynamics*.”

## Reviewer C

16. Supplementary information. Please add a table of contents, including clear headings and subheadings, to improve the readability of the 74 pages.

## Authors's response

Reply #16: As requested, we have now added a detailed table of contents at the beginning of the supplementary document to improve clarity and ease of navigation to contents of interest. This now also includes clear headings and subheadings that correspond to the major sections and subsections. We reproduce this below.

Table of Contents

**Table d67e8444:** 

Intra-class correlations	1
Model parameters by group	1
Descriptive task measures	1
Intra-class correlations for post-task survey questions	5
Relationship between model parameters and other measures at follow-up	5
Demographic variables	5
Descriptive task measuresy	6
Post-task surveys	6
Group differences in model parameters when only including participants who returned for follow-up	7
Group differences in post-task self-report questionnaire ratings	12
Group differences in the relationship between response times and chosen runway positions	13
Group differences in the relationship between response times and model parameters	14
Supplementary Figures	16
Supplementary Figure 1	16
Supplementary Figure 2	18
Supplementary Figure 3	19
Supplementary Figure 4	20
Supplementary Figure 5	21
Supplementary Figure 6	22
Supplementary Figure 7	23
Supplementary Figure 8	24
Supplementary Figure 9	25
Supplementary Figure 10	25
Supplementary Tables	26
Supplementary Table 1. Comorbidity table for participants who returned for follow-up	26
Supplementary Table 2. Differences in participant characteristics at baseline for those who did vs. did not return for follow-up	26
Supplementary Table 3. Baseline symptom and demographic characteristics by group for participants who did vs. did not return for follow-up	26
Supplementary Table 4. Group-wise differences in participant characteristics at 1-year follow-up in the exploratory and confirmatory sample	27
Supplementary Table 5. Group-wise intra-class correlations for clinical measures at baseline and 1-year follow-up	27
Supplementary Table 6. Group-wise intra-class correlations for post-task self-report questionnaire items at baseline and 1-year follow-up	29
Supplementary Table 7. Group-wise post-task self-report questionnaire items at baseline and 1-year follow-up, and correlations with computational model parameters at follow-up	30
Supplementary Table 8. Results of linear mixed effects models predicting *DU* and *EC* in data including all baseline participants, when accounting for effects of group and time	32
Supplementary Table 9. Results of linear mixed effects models predicting *DU* and *EC* in data including all baseline participants, when accounting for effects of group, time, age, sex, and WRAT scores	32
Supplementary Table 10. Summary statistics for response times (Mean (SD)) at baseline and follow-up	34
Supplementary Table 11. Summary statistics for average chosen runway position (Mean (SD)) at baseline and follow-up	35
Supplementary Table 12. Summary statistics for variability (SD) in chosen runway position (Mean (SD)) at baseline and follow-up	36
Supplementary Table 13. Results of linear mixed effects models predicting response times (RTs) in participants who returned for follow-up, when accounting for effects of group, time, and their interaction	37
Supplementary Table 14. Results of linear mixed effects models predicting response times (RTs) in participants who returned for follow-up, when accounting for effects of group, time, age, and sex	38
Supplementary Table 15. Results of linear mixed effects models predicting response times (RTs) in participants who returned for follow-up, when accounting for effects of group, time, age, and sex	39
Supplementary Table 16. Results of linear mixed effects models predicting average chosen runway position in participants who returned for follow-up, when accounting for effects of group, time, and their interaction	42
Supplementary Table 17. Results of linear mixed effects models predicting average chosen runway position in participants who returned for follow-up, when accounting for effects of group, time, age, and sex	43
Supplementary Table 18. Results of linear mixed effects models predicting average chosen runway position in participants who returned for follow-up, when accounting for effects of group, time, age, sex, and WRAT scores	45
Supplementary Table 19. Results of linear mixed effects models predicting choice variability in participants who returned for follow-up, when accounting for effects of group, time, and their interaction	47
Supplementary Table 20. Results of linear mixed effects models predicting choice variability in participants who returned for follow-up, when accounting for effects of group, time, age, and sex	49
Supplementary Table 21. Results of linear mixed effects models predicting choice variability in participants who returned for follow-up, when accounting for effects of group, time, age, sex and WRAT scores	51
Supplementary Table 22. Results of linear mixed effects models predicting self-report questionnaire items in participants who returned for follow-up, when accounting for effects of group, time, and their interaction	53
Supplementary Table 23. Results of linear mixed effects models predicting self-report questionnaire items in participants who returned for follow-up, when accounting for effects of group, time, age, and sex	54
Supplementary Table 24. Results of linear mixed effects models predicting self-report questionnaire items in participants who returned for follow-up, when accounting for effects of group, time age, sex, and WRAT scores	55
Supplementary Table 25. Summary statistics for model parameters (Mean (SD)) for each sub-diagnosis at baseline and follow-up in the exploratory and the confirmatory samples	57
Supplementary Table 26. Results of linear mixed effects models predicting *DU* when accounting for effects of group, time, and their interaction	58
Supplementary Table 27. Results of linear mixed effects models predicting *DU* when accounting for effects of group, time, age, and sex	59
Supplementary Table 28. Results of linear mixed effects models predicting *EC* when accounting for effects of group, time, and their interaction	62
Supplementary Table 29. Results of linear mixed effects models predicting *EC* when accounting for effects of group, time, age, and sex	64
Supplementary Table 30. Simulation-based post-hoc power analyses to detect group differences between HCs and clinical subgroups	69
Supplementary Table 31. Performance metrics for predictive categorization of individuals with affective disorders (no comorbid SUDs) and those with SUDs (but no comorbid affective disorders)	69
Supplementary Table 32. Performance metrics for predictive categorization of individuals with and without SUDs	70
Supplementary Table 33. Performance metrics for predictive categorization of individuals with and without affective disorders	71
Supplementary Table 34. Performance metrics for predictive categorization of individuals with and without comorbid affective and substance use disorders	71

## Reviewer C

17. To improve readability, add page numbers to the main manuscript and supplementary materials.

## Author’s response

Reply #17: We have now added page numbers to the main manuscript and supplementary materials, as requested.

Reviewer D:

Recommendation: Accept Submission

Overview

Thank you for the opportunity to review the manuscript titled “Computational Mechanisms of Approach-Avoidance Conflict Predictively Differentiate Between Affective and Substance Use Disorders”. After carefully reviewing the manuscript, I am confident in the rigor of its implementation and its contribution to the field and would recommend to publish it almost as-is. Below, I provide a brief summary of the strengths of each section and provide some minor suggestions.

Overall Reply: We thank the reviewer for their time in reviewing the paper and identifying ways to improve the quality of our submission. We have addressed each concern in more detail below.

Introduction

The study examined computational markers of decision-making in an approach-avoidance conflict task in a large, longitudinal sample. Of specific interest were mechanisms of decision uncertainty and emotion conflict, both of which have important transdiagnostic implications in clinical settings. This study replicates an earlier study of computational approach-avoidance mechanisms in an exploratory sample, using a confirmatory sample. All planned analyses were pre-registered, and the reported results are consistent with the pre-registration.

The introduction nicely summarizes the state of the art in the field, and opens the question of transdiagnostic processes in psychiatry. It concisely summarizes the value of computational models of cognitive processes and its challenges.

- Here, I would suggest adding some more references to prior work into the temporal stability of computational markers of decision-making. Some research points towards rather poor psychometric properties of computational markers and quantities – e.g., Karvelis, Paulus & Diaconescu (2023). In contrast, see Goodwin, Hester & Garrido (2024) who demonstrate good ICC for Bayesian belief-updating parameters.

The introduction also locates the present work in the larger context of the Tulsa 1000 study and identifies the present study as a replication with some additional, more granular analyses conducted on both samples.

## Peer-review-recommendation

Resubmit for Review

---

## [Reviewer Report · Peer Review History Round 1, Anna-Lena Eckert]

## Reviewer D

Thank you for the opportunity to review the manuscript titled “Computational Mechanisms of Approach-Avoidance Conflict Predictively Differentiate Between Affective and Substance Use Disorders”. After carefully reviewing the manuscript, I am confident in the rigor of its implementation and its contribution to the field and would recommend to publish it almost as-is. Below, I provide a brief summary of the strengths of each section and provide some minor suggestions.

The study examined computational markers of decision-making in an approach-avoidance conflict task in a large, longitudinal sample. Of specific interest were mechanisms of decision uncertainty and emotion conflict, both of which have important transdiagnostic implications in clinical settings. This study replicates an earlier study of computational approach-avoidance mechanisms in an exploratory sample, using a confirmatory sample. All planned analyses were pre-registered, and the reported results are consistent with the pre-registration.

The introduction nicely summarizes the state of the art in the field, and opens the question of transdiagnostic processes in psychiatry. It concisely summarizes the value of computational models of cognitive processes and its challenges.

- Here, I would suggest adding some more references to prior work into the temporal stability of computational markers of decision-making. Some research points towards rather poor psychometric properties of computational markers and quantities – e.g., Karvelis, Paulus & Diaconescu (2023). In contrast, see Goodwin, Hester & Garrido (2024) who demonstrate good ICC for Bayesian belief-updating parameters.

The introduction also locates the present work in the larger context of the Tulsa 1000 study and identifies the present study as a replication with some additional, more granular analyses conducted on both samples.

## Author’s response

Reply #18: We agree it would be helpful to incorporate additional references to prior work addressing (in)stability of computational markers. We have therefore attempted to incorporate the work suggested by the reviewer. An example of relevant revised language is reproduced below:

Pg. 4 - *“More recently, computational modeling approaches have highlighted cognitive processes, such as suboptimal reward valuation, uncertainty, or inference, that may contribute to maladaptive AAC behavior. However, the clinical utility of these findings remains limited by sparse evidence of replicability and longitudinal stability of computational markers, as well as indications of variability in test-retest reliability (ranging from poor-to-excellent) depending on the task in question (Brown et al., 2020; Chung et al., 2017; Enkavi et al., 2019; Karvelis et al., 2023; Moutoussis et al., 2019; Price et al., 2019; Shahar et al., 2019). This limitation partly arises from the prevalent use of cross-sectional study designs, though more recent shifts toward longitudinal methodologies have facilitated identification of computational markers demonstrating good reliability (Goodwin et al., 2024). However, these investigations have predominantly focused on relatively short intervals spanning days to weeks. Investigations focusing on long-term stability will be crucial for progress in computational psychiatry and its continued efforts to identify underlying mechanisms and predictors of treatment outcomes”*

## Reviewer D

Regarding the methods, I think that this study is exemplary in the rich dataset used, the computational model, and the machine learning methods implemented. Since the authors aim to replicate a prior study, analyses techniques follow previously published work very closely. The dataset of the Tulsa 1000 study is rather unique with respect to the wealth of data and the sample size, making it a very valuable resource for both descriptive and predictive computational analyses, and for the investigation of psychometric properties of computational markers as done here.

- Minor clarification question: in the “Participants” Section, the scores (e.g. PHQ ≥10) represents the inclusion criteria after screening, correct? It might be good to clarify in the text.

The AAC task is a classical task in the field and the authors use a well-validated set of stimuli; all of which have been described previously. Regardless, I would wish for some additional information on the task here:

- Clarification question on the task design: Moving the avatar towards a cloud, participants are instructed that it will increase the probability of seeing a negative affect stimulus by 10% per step. If I am e.g. on step 8, I should expect a negative stimulus pair in 80% of trials. What precisely happens in the other 20% of trials – no stimulus, correct? Same with the reward; on step 8, I would risk not receiving the reward in 20% of trials, correct? I think it would be important to mention this.

- Were participants aware of their global reward count?

- As a reader I would appreciate a brief description of the sounds used, even though it was published elsewhere I think it would improve the clarity of the task description here to mention some of its basic features (e.g., naturalistic or artificial? Loudness?)

- If the collected rewards did not influence their payout in the end, how did you make sure that AAC is triggered reliably?

## Author’s response

Reply #19: We appreciate how more details about the task would be helpful. To address this, the task description and the figure legend have now been updated to include the further details mentioned by the reviewer. We provide answers to each of the reviewer’s question below in the order they were asked:

Task design: In the reviewer’s example, they correctly understand that, if at step 8, they should receive the negative stimulus pair on 80% of trials. However, they are incorrect that on 20% of trials nothing would happen. Instead, in these 20% of cases they would receive the alternative positive outcome (represented on the side of the runway that they chose to move away from). We have now explicitly clarified this point in the legend of Figure 1 to ensure there is no ambiguity regarding trial outcomes, which we reproduce below.Global Reward Count: Participants were aware of the overall number of points they had won at each point during the task. Namely, both the immediate and cumulative rewards were displayed on-screen during the reward phase of each trial. This detail has now also been included in the Figure 1 legend, as reproduced below.
Figure 1**The approach-avoidance conflict (AAC) task. *Bottom:*** Each trial is divided into a decision phase, an affective stimulus phase, and a reward phase. Trials are separated by a variable intertrial fixation time. ***Top:*** During the decision phase, participants choose to move an avatar to one of nine positions on a runway. Pictures are presented on each side of the runway, indicating the types of stimuli that could be presented during the affective stimulus and reward phases. The sun and cloud images represented potential positive and negative affective stimuli, respectively (each being an image–sound combination). The height of the red fill in a rectangle signified the number of points that would be received in the reward phase (ranging from 0 to 6 points). Participants were instructed that the final position of the avatar determined the probability of each of these outcomes occurring (in increments of 10%, from 90% to 10% with each step away from the associated stimulus indicator images). All choices therefore resulted in a probabilistic outcome. For example, if a participant chose the highest probability option on a given side (i.e., the runway position closest to their preferred outcome), there was a 90% chance that the preferred outcome would be presented. However, there was still a 10% chance that the non-preferred outcome associated with the alternative side of the runway would be presented instead. In the CONF6 condition above, for instance, the preferred outcome might be the combination of a negative affective stimulus and 6 points, while the alternative outcome would be a pleasant affective stimulus associated with no points. At the end of each trial (i.e., in the reward phase), participants were informed of the points won on that trial (i.e., including when 0 points were earned) as well as the total number of points they had acquired in the task thus far. The five trial types and associated probabilities of each outcome at each runway position are also shown above. The task consisted of 60 trials, with 12 of each of the five trial types.AAC task trial types and structure
Sound description: We agree that a brief description of the aversive sounds would improve clarity. The auditory stimuli were either pleasant (e.g., wind bells, pleasant laughter) or aversive (e.g., screaming noises, nails on a blackboard) naturalistic sounds, sampled from the International Affective Digitized Sounds (IADS; Lang & Bradley, 1999), and other freely available audio files. We have now added language describing task in more detail, including the nature of the sounds. Here is an excerpt exemplifying this revised language (see final few sentences):Pg. 8 - *“The AAC task (Figure 1) used here has been described in detail in our previous work (Aupperle et al., 2015; Aupperle et al., 2011; Smith et al., 2021b). Briefly, a picture of an avatar standing above a runway, at its starting position, was presented on each trial. Participants were asked to move the avatar to one of the nine positions on the runway, toward the cues presented on either side. The cue consisted of an image – sun or cloud – representing a positive or negative affective image-sound pair, respectively, that would be shown with higher probability as the avatar moved closer to the associated side (detailed below). There was also a rectangular bar on each side, where the height of the red fill represented reward points associated with the linked image-sound pair. There were five trial types across 60 trials (12 each), defined by the cues presented: avoid-threat (AV), approach-reward (APP), and three levels of conflict trials (CONF2, CONF4, CONF6). In AV trials, the cloud and sun images were presented on opposing sides, with 0 reward points associated with either. APP trials provided positive affective stimuli on both sides, where one side was associated with 0 points and the other with 2 points. Lastly, in conflict trials, cues presented always included the sun with 0 reward points and the cloud with levels of 2 (CONF2), 4 (CONF4), and 6 (CONF6) reward points. These reward points did not lead to additional monetary compensation. The affective images and sounds were sampled from the International Affective Picture System (IAPS; Lang et al., 2008), International Affective Digitized Sounds (IADS; Lang & Bradley, 1999), and other freely available audio files (refer to Aupperle et al. (2015) and Chrysikou et al. (2017)). Some examples of visual stimuli include images depicting violence or suffering, scenic natural environments, and people displaying positive emotions (e.g., smiling or laughing) or engaging in joyful activities. Examples of associated auditory stimuli include screaming noises, the sound of nails on a blackboard, wind bells, and pleasant laughter, to name a few.”*Non-monetary AAC Operationalization: We agree it is important to be sure that, despite lack of monetary reward, the task points and affective stimuli nonetheless elicit AAC. In our view, the patterns of behavior we observe appear to support AAC, in that individuals will approach for some amounts of points but avoid for others; or they will approach when only points are on offer but avoid whenever negative stimuli are possible. We similarly observe indications of AAC in post-task questions probing levels of decision difficulty, anxiety, and approach-avoidance motivation. As shown in Table 5 and Supplementary Figure 3 (both reproduced below), our computational behavioral measures from the task also correlate highly with these self-reports. This pattern, in combination with the large number of prior studies that have successfully induced AAC with this task, make us confident that AAC was elicited in our study.Pg. 24 - *“Relationships between model parameters and post-task survey questions were also in expected directions, as detailed in Table 5. In particular, there were significant positive associations between EC and both self-reported anxiety (Q2) and avoidance motivation (Q5) during the task, and a positive association between DU and greater self-reported decision difficulty (Q3) on the task (Supplementary Figure 3). As the model was not fit to these RT and self-report measures, this provides external support for its validity*.”
Table 5Post-task self-report questionnaire items at baseline and follow-up, and correlations with computational model parameters at follow-up.

POST-TASK SELF-REPORT QUESTIONS (LIKERT SCALE: 1 = NOT AT ALL; 7 = VERY MUCH)MEAN (SD) BASELINEMEAN (SD) 1-YEAR FOLLOW-UP (N = 287)EMOTION CONFLICT PARAMETER (*EC*)DECISION UNCERTAINTY PARAMETER (*DU*)

ALL PARTICIPANTS(N = 480)RETURNED FOR FOLLOW-UP(N = 287)

Q1. I found the positive pictures enjoyable5.02 (1.56)5.17 (1.56)4.97 (1.59)0.06–0.04

Q2. The negative pictures made me feel anxious or uncomfortable4.01 (1.97)3.98 (1.99)4.00 (1.95)
**0.34*****

**0.06**

Q3. I often found it difficult to decide which outcome I wanted2.32 (1.71)2.26 (1.68)2.07 (1.60)0.04
**0.42*****

Q4. I always tried to move all the way towards the outcome with the largest reward points4.86 (2.35)4.84 (2.38)^┼^4.98 (2.44)
**–0.77*****

**–0.46*****

Q5. I always tried to move all the way away from the outcome with the negative picture/sounds2.86 (2.15)2.92 (2.19)2.93 (2.32)
**0.72*****

**0.31*****

Q6. When a negative picture and sound were displayed, I kept my eyes open and looked at the picture5.43 (1.89)5.39 (1.89)^┼^5.29 (1.97)
**–0.42*****

**–0.21*****

Q7. When a negative picture and sound were displayed, I tried to think about something unrelated to the picture to distract myself2.84 (1.90)2.93 (1.95)2.95 (1.97)
**0.32*****
0.04

Q8. When a negative picture and sound were displayed, I tried other strategies to manage emotions triggered by the pictures3.04 (1.91)3.11 (1.90)3.22 (1.93)
**0.34*****

**0.05**

**Note:**
^┼^*p* < 0.05, ^┼┼^*p* < 0.01, ^┼┼┼^*p* < 0.001 (pre-post differences); **p* < 0.05, ***p* < 0.01 ****p* < 0.001 (correlations at follow-up). Results that were statistically significant in the exploratory sample are highlighted in bold.

**Supplementary Figure 3 SF3:** Association between model parameters (top: DU and bottom: EC) and (left) select self-report questionnaire items (Q2 and Q3 in Table 5, main text) and (right) response time (RTs) in approach (top) and avoid (bottom) conditions.

## Reviewer D

The authors use a POMDP with 2 parameters, Decision Uncertainty (DU) and Emotion Conflict (EC), which were obtained by fitting the model to participant data using the Variational Laplace approach. They also investigated choice uncertainty (RT-based), checking also the influence of RT on chosen position and model parameters. Sample differences at baseline vs. follow-up were measured using t-tests, regression and Bayes Factors. For within-subject stability, ICC were reported, which is robust also for unequal group sizes. Partial correlations are used to control for baseline participant characteristics. I found the stacked ensemble approach to classification very interesting and implemented rigorously. The methods section is written very clearly and all analyses are well-motivated and follow pre-registered, established and validated procedures.

## Author’s response

Reply: We thank the reviewer for these positive comments and are glad to hear they approved of these aspects of our approach.

## Reviewer D

Results. The authors do find some evidence of selective drop-out (i.e., based on sex, pre-morbid cognitive capacity, drug abuse), which did not affect their modelling results; and was further controlled for with the ITT principle. The model accurately predicts behavior in over 80% of trials, which is a good performance, especially given the fact it has only 2 parameters. The results are constantly brought into perspective by comparing them to the exploratory sample results, which allows very interesting longitudinal conclusions. All analyses are reported concisely, but with sufficient detail, and comprehensive supplementary results are provided. The predictive classification results are very interesting.

- Figure 3: could the authors please add a legend for the spaghetti plots? This would improve interpretability.

## Author’s response

Reply #20: We appreciate the reviewer bringing this to our attention. We have now updated the figure legend to address this as follows (now Figure 4):

**Figure 4 d67e9127:** **Pre-registered tests of group, time, and sex effects on model parameters**. Spaghetti plots (left) show changes from baseline to follow-up and therefore only include participants who returned for the follow-up visit. Bar plots (right) include all baseline participants, including those who did not return for follow-up. These plots illustrate individual differences in the stability of parameter values over time. Here, thick lines within the spaghetti plots indicate group means and lighter lines indicate individual values. Shaded areas around the line for each group mean reflect the associated standard errors. Sex comparisons in the lower bar graphs illustrate how observed group differences in *EC* were mainly driven by females. **p* < 0.05, ***p* < 0.01, ****p* < 0.001.

## Reviewer D

The discussion section concisely summarizes and explores the implications of the most important findings: 1) greater EC values in HC, 2) reduced DU values over time, 3) good predictive classification of DEP/ANX or SUD based on model parameters and few demographic characteristics. It includes a discussion of the study’s limitations and how they were addressed in the presented analyses.

Minor issues and typos:

- Please use continuous page & line numbers in the future for easier reference.

## Author’s response

Reply #21: We have now added page numbers to the main manuscript and supplementary materials, as suggested.

## Reviewer D

- Introduction: “,which may contribute to maladaptive AAC behavior c.” (typo)

## Author’s response

Reply #22: Thank you for bringing this to our attention. This typo has been fixed.

Pg. 4 - “*More recently, computational modeling approaches have highlighted cognitive processes, such as suboptimal reward valuation, uncertainty, or inference, that may contribute to maladaptive AAC behavior*.”